# Multi-Step Alignment as Markov Games: An Optimistic Online Mirror Descent Approach with Convergence Guarantees

**Yongtao Wu**[*†]    **Luca Viano**[*†]    **Kimon Antonakopoulos**[†]    **Yihang Chen**[‡]
**Zhenyu Zhu**[†]    **Quanquan Gu** [‡§]    **Volkan Cevher** [†§]

**Reviewed on OpenReview:** `https://openreview.net/forum?id=ZWZKaqZCy0`

## Abstract

Reinforcement Learning from Human Feedback (RLHF) has been highly successful in aligning large language models with human preferences. While prevalent methods like DPO have demonstrated strong performance, they frame interactions with the language model as a bandit problem, which limits their applicability in real-world scenarios where multi-turn conversations are common. Additionally, DPO relies on the Bradley-Terry model assumption, which does not adequately capture the non-transitive nature of human preferences. In this paper, we address these challenges by modeling the alignment problem as a two-player constant-sum Markov game, where each player seeks to maximize their winning rate against the other across all steps of the conversation. Our approach Optimistic Multi-step Preference Optimization (`OMPO`) is built upon the optimistic online mirror descent algorithm (Rakhlin & Sridharan, 2013; Joulani et al., 2017). Theoretically, we provide a rigorous analysis for the convergence of `OMPO` and show that `OMPO` requires $\mathcal{O}(\epsilon^{-1})$ policy updates to converge to an $\epsilon$-approximate Nash equilibrium. We also validate the effectiveness of our method on multi-turn conversations dataset and math reasoning dataset.

## 1 Introduction

In recent years, the integration of large language models (LLMs) (Brown et al., 2020; Achiam et al., 2023; Team et al., 2023; Dubey et al., 2024) into various applications has highlighted the need for advanced preference alignment methods (Ziegler et al., 2019; Stiennon et al., 2020; Bai et al., 2022; Ouyang et al., 2022; Rafailov et al., 2023; Guo et al., 2025). As models increasingly engage in complex decision making or reasoning scenarios, the ability to align their outputs with user preferences requires a learning algorithm that satisfies the following desiderata.

- **Desiderata 1: Multi-step learning with intermediate preference signal.** In multi-round conversations, alignment must occur at each turn to meet user needs. Similarly, in mathematical reasoning with chain-of-thought prompting, step-by-step validation is essential to ensure accuracy in the final result. Unfortunately, most existing works on reinforcement learning from human feedback (RLHF) focus on one-step preference (Rafailov et al., 2023; Meng et al., 2024; Munos et al., 2024; Azar et al., 2024; Zhang et al., 2024; Wu et al., 2025). In addition, most of the multi-step works (Wang et al., 2023; Shani et al., 2024; Swamy et al., 2024) assume that the preferences are revealed only at the terminal state, neglecting intermediate preferences.

- **Desiderata 2: General preferences.** The learning algorithm can handle general, non-transitive preference models, bypassing the Bradley-Terry assumption (Bradley & Terry, 1952), which assigns a score for each answer based on its preference. This assumption of the model cannot capture the non-transitive preference, which is often observed in the averaged human preferences from the population (Tversky, 1969; Gardner, 1970).

---

† EPFL ‡ UCLA * Equal contribution § Equal mentorship.    Correspondence to: yongtao.wu@epfl.ch

Table 1: Comparison between the literature of learning from a general preference oracle, which may violate the Bradley–Terry assumption. $^\dagger$ denotes that this rate applies for convergences to the Nash equilibrium (NE) of the regularized game, obtained by adding a penalty in the form $\alpha D(\cdot, \pi_{\mathrm{ref}})$, where $D$ denotes the KL divergence. IPS stands for intermediate preference signal. $^\star$ denotes that the last iterate convergence is asymptotic only. Detailed related work can be found in Appx. B.

| Algorithm | IPS | Multi Step | Updates for $\epsilon$-NE | Without $\alpha$-strong convexity | $\alpha$ | Last Iterate Guarantees |
|---|---|---|---|---|---|---|
| SPPO Wu et al. (2025) | ✗ | ✗ | $\mathcal{O}(\varepsilon^{-2})$ | ✓ | – | ✗ |
| SPO Swamy et al. (2024) | ✗ | ✓ | $\mathcal{O}(\varepsilon^{-2})$ | ✓ | – | ✗ |
| MTPO Shani et al. (2024) | ✗ | ✓ | $\mathcal{O}(\alpha^{-2}\varepsilon^{-1})^\dagger$ | ✗ | 0.0025 | ✓ |
| Nash-MD Munos et al. (2024) | ✗ | ✗ | $\mathcal{O}(\alpha^{-2}\varepsilon^{-1})^\dagger$ | ✗ | 0.008 | ✓ |
| EGPO Zhou et al. (2025) | ✗ | ✗ | $\mathcal{O}(|\mathcal{Y}|\varepsilon^{-1})$ | ✓ | – | ✓ |
| ONPO Zhang et al. (2025c) | ✗ | ✗ | $\mathcal{O}(\varepsilon^{-1})$ | ✓ | – | ✗ |
| MMD Wang et al. (2024) | ✗ | ✗ | asymptotic | ✓ | – | ✓$^\star$ |
| **OMPO (Ours)** | ✓ | ✓ | $\mathcal{O}(\varepsilon^{-1})$ | ✓ | – | ✓$^\star$ |

- **Desiderata 3: Convergence guarantees.** It has reliable and robust convergence guarantees in the multi-turn setting. Recent work Shani et al. (2024) considers an $\alpha$-regularized preference problem and exploits its strong convexity to derive convergence bounds. Unfortunately, these bounds are not very informative when the regularization strength $\alpha$ tends to 0. It remains open to prove a convergence rate which does not deteriorate for vanishing $\alpha$. Moreover, a non vacuous upper bound on the number of policies updates should depend on the number of sentences $|\mathcal{Y}|$ at most logarithmically.

In this paper, we present the first algorithm achieving the three desiderata at once by formulating multi-step general preference optimization within the framework of two-player Markov games (Shapley, 1953). In a two-player Markov game, each player seeks to maximize their winning rate against the other across all steps of the conversation.

Moreover, for the multi-turn learning from preference, it is enough to consider a Markov game where each player has their own state and the transition dynamics do not depend on the state of the other player. Under this setting, we can leverage techniques from the linear programming literature in Markov decision processes Manne (1960) to formulate the multi-step problem as a bilinear problem over the space of the occupancy measures.

We then apply the optimistic online mirror descent algorithm (Rakhlin & Sridharan, 2013; Joulani et al., 2017) to obtain fast convergence guarantees. In particular, we show that it is possible to find an $\varepsilon$-Nash equilibrium of this game in $\mathcal{O}(\varepsilon^{-1})$ gradients updates. Moreover, leveraging Lagrangian duality, we show that the optimistic online mirror descent update can be implemented in a projection free manner, making it suitable for a practical implementation.

We name the derived algorithm Optimistic Multi-step Preference Optimization (`OMPO`). Numerical results demonstrate that `OMPO` attains considerable improvements on multi-turn conversation datasets and math reasoning datasets. Our contribution is compared to the recent literature on the same topic in Tab. 1.

## 2 Problem setting: Multi-step RLHF as two-player Markov games

### 2.1 Notation

We define the prompt to the language model as $x$ and the answer from the language model as $a$. For a multi-turn conversation with turn $H$, the prompts and the answers are denoted by $x_h$ and $a_h, \forall h \in [H]$. The concatenation of a prompt $x$ and an answer $a$ is denoted by $[x, a]$ and can be generalized to the concatenation of multiple prompts and answers, e.g., $[x_1, a_1, \ldots, x_H, a_H]$.

For any two prompt action sequences, e.g., $y = [x_1, a_1, \ldots, x_H, a_H]$ and $y' = [x'_1, a'_1, \ldots, x'_H, a'_H]$, we define a preference oracle as $o(y \succ y') \in \{0, 1\}$, which can provide preference feedback with 0-1 scores, where 1 means the conversation $y$ is preferred and 0 otherwise. We denote $\mathbb{P}(y \succ y') = \mathbb{E}[o(y \succ y')]$ as the probability that the conversation $y$ is preferred over $y'$. Moreover, we have $\mathbb{P}(y \succ y') = 1 - \mathbb{P}(y' \succ y)$.

An autoregressive language model is denoted by $\pi(a|x)$, which receives input $x$ and generates answer $a$. We denote the KL divergence of two probability distributions $p$ and $q$ by $D(p, q)$. The Bregman Divergences between two points are denoted by $\mathbb{D}(p, q)$. The sigmoid function is defined by $\sigma(z) := \frac{1}{1+e^{-z}}$. Moreover, we use capital letters to denote random variables: for example, $s$ denotes a specific state, while $S$ represents a state sampled from a certain distribution. We use $\|\cdot\|_\infty$ to denote the $\ell_\infty$-norm. Detailed definitions for the notations are summarized in Appx. A.

## 2.2 Problem formulation of multi-step RLHF

In this section, we introduce the problem setting for multi-step RLHF. Specifically, we can cast the multi-step alignment process as an episodic finite-horizon Markov Decision Process (MDP). An MDP is a tuple $\mathcal{M} = (\mathcal{S}, \mathcal{A}, f, r, \nu_1, H)$, where $\mathcal{S}$ is the state space, $\mathcal{A}$ is the action space, $H$ is the horizon (total steps), the initial state distribution $\nu_1$ is a distribution over the state space $\mathcal{S}$. A potentially non-stationary policy $\pi : \mathcal{A} \times [H] \to \Delta_{\mathcal{A}}$ is a mapping from states (sentences) and stages to distribution over actions. We define the policy set as $\Pi$.

Sampling an episode is done according to the following protocol. At the initial step, we sample the prompt $X_1 \sim \nu_1$ and define the initial state equal to the prompt itself, i.e., $S_1 = X_1$. For each step $h$, a new action $A_h \sim \pi_h(\cdot|S_h)$ is sampled from the policy and the next prompt is sampled according to the transition function $f$, that is $X_{h+1} \sim f(\cdot|S_h, A_h)$. The next state is then constructed deterministically as $S_{h+1} = [S_h, A_h, X_{h+1}]$ by using the concatenation operator between sentences. An important consequence is that the MDP is tree-structured and that each state can be reached only from a single initial state. The episodes end after $H$ steps.

Our setting covers a number of alignment problems, and we list some examples below.

**Example 1** (Single-step alignment)**.** *In single-step alignment, a language model receives one prompt and outputs one answer. Our framework covers the single-step alignment by dissecting the answer into single tokens. Specifically, we set $X_1$ as the prompt, $X_2, \ldots, X_{H+1}$ as empty sentences, and the answer $A_h$ at each turn consists of only one token. Then the horizon $H$ is the number of tokens in the answer. The transition between each state is deterministic.*

**Example 2** (Chain-of-thought reasoning alignment)**.** *In the chain-of-thought reasoning, the horizon $H$ denotes the number of reasoning steps, where $X_1$ is the initial prompt and $X_2, \ldots, X_{H+1}$ are empty. Each $A_h$ corresponds to a reasoning step. The transition between each state is deterministic.*

**Example 3** (Multi-turn conversation alignment)**.** *In multi-turn conversation, the horizon $H$ denotes the total number of turns in the conversation. In the $h$-th turn, $X_h$ is the prompt, and $A_h$ is the answer. The prompt in the terminal state, $X_{H+1}$, is an empty sentence. The transition between each state can be deterministic or stochastic.*

Next, we define the pair-wise reward function of two state-action pairs $(s, a) \in \mathcal{S} \times \mathcal{A}$ and $(s', a') \in \mathcal{S} \times \mathcal{A}$ as the preference of two trajectories: $r(s, a, s', a') = \mathbb{P}([s, a] \succ [s', a'])$. Notice that, by definition, we have that $\|r\|_\infty \leq 1$.

We aim to identify the Nash equilibrium of the following two-player symmetric constant-sum Markov game[1]:

$$(\pi^*, \pi^*) = \arg\max_{\pi \in \Pi} \min_{\pi' \in \Pi} \mathbb{E}_{S_1 \sim \nu_1, \pi, \pi'} \left[ \sum_{h=1}^{H} r(S_h, A_h, S'_h, A'_h) \right], \tag{Game}$$

---

[1]In a two-player constant-sum Markov game, a Nash equilibrium is defined as $(\pi_1^\star, \pi_2^\star) = \arg\max_\pi \min_{\pi'} \mathbb{E}_{S_1 \sim \nu_1} V^{\pi, \pi'}(S_1, S_1)$, where $\pi_1^\star$ and $\pi_2^\star$ do not necessarily coincide in general. However, in the symmetric game setting (our setting), the two players face identical action and state spaces and share the same best-response structure. In this case, the equilibrium policies coincide so that we write the equilibrium compactly as $(\pi^\star, \pi^\star)$. The same convention applies to the occupancy measures $(d^\star, d^\star)$.

where the two state action sequences are generated with the above protocol $\{(S_h, A_h)\}_{h=1}^H$ and $\{(S_h', A_h')\}_{h=1}^H$. We enforce $S_1' = S_1$ to guarantee the two agents start from the same prompt.

For the reader's convenience, we elaborate further on Game in Appx. G, discussing its interpretation in terms of the $\max \min$ operator, the role of the input $X_1$, the time horizon $H$, the availability of $\mathbb{P}$, and minimal examples that illustrate the advantages of general preferences and intermediate rewards.

## 2.3 Useful facts in Markov games

Next, we present some additional quantities and notation which help in dealing with Markov games. We define the *pair-wise* state and state action value functions as follows

$$V_h^{\pi,\pi'}(s, s') = \mathbb{E}_{\pi,\pi'}\Big[ \sum_{\tau=h}^H r(S_\tau, A_\tau, S_\tau', A_\tau') | S_h = s, S_h' = s'\Big],$$

$$Q_h^{\pi,\pi'}(s, a, s', a') = \mathbb{E}_{\pi,\pi'}\Big[ \sum_{\tau=h}^H r(S_\tau, A_\tau, S_\tau', A_\tau') | S_h = s, S_h' = s', A_h = a, A_h' = a'\Big],$$

where $A_\tau \sim \pi_\tau(\cdot|S_\tau)$, $A_\tau' \sim \pi_\tau'(\cdot|S_\tau')$, $S_{\tau+1} \sim f(\cdot|S_\tau, A_\tau)$, and $S_{\tau+1}' \sim f(\cdot|S_\tau', A_\tau')$. We will often denote $V_1^{\pi,\pi'}$ without the subscript, i.e., as $V^{\pi,\pi'}$. Notice that since the reward function is bounded by 1, we have that $\|V_h^{\pi,\pi'}\|_\infty \leq H$ and $\|Q_h^{\pi,\pi'}\|_\infty \leq H$ for all $h \in [H]$. Moreover, notice that we consider potentially non-stationary policies. In particular, $\pi_h$ denotes the probability of choosing actions at stage $h$ and $\pi = (\pi_1, \dots, \pi_H)$ denotes the global policy that samples actions according to $\pi_h$ at stage $h$.

**Remark 1.** *Readers might be surprised on reading a double state dependence in the definition of the state value function. Indeed, in standard literature of two-player Markov games, the tuple $s, s'$ is considered as a joint common state. Therefore, the agents generate the next actions sampling from policies conditioned on the joint state ($\pi_h(\cdot|s_h, s_h')$). This protocol is not suitable for the conversation task in which each agent (LLM) should generate the next action conditioned only on its own state (conversation up to stage h). This motivates our choice of not representing $s, s'$ as a common joint state.*

Having introduced the value functions, we can rewrite Game in terms of state value functions as follows:

$$(\pi^*, \pi^*) = \arg\max_{\pi \in \Pi} \min_{\pi' \in \Pi} \mathbb{E}\Big[ \sum_{h=1}^H r(S_h, A_h, S_h', A_h')\Big] = \arg\max_\pi \min_{\pi'} \mathbb{E}_{S_1 \sim \nu_1} V^{\pi,\pi'}(S_1, S_1). \tag{1}$$

Moreover, we will use the following compact inner product notation $\mathbb{E}_{S_1 \sim \nu_1} V^{\pi,\pi'}(S_1, S_1) = \left\langle \nu_1, V^{\pi,\pi'} \right\rangle$. Given the above notation, we can formalize our objective. We look for a policy $\pi$ satisfying the following definition of approximate equilibrium.

**Definition 1** ($\epsilon$-approximate Nash equilibrium(Orabona, 2019; Daskalakis et al., 2020; Sayin et al., 2021)). *A policy $\pi$ is said to be an approximate Nash equilibrium if it holds that:*

$$\left\langle \nu_1, V^{\pi,\pi} \right\rangle - \min_{\bar{\pi} \in \Pi} \left\langle \nu_1, V^{\pi,\bar{\pi}} \right\rangle \leq \epsilon, \quad and \quad \max_{\bar{\pi} \in \Pi} \left\langle \nu_1, V^{\bar{\pi},\pi} \right\rangle - \left\langle \nu_1, V^{\pi,\pi} \right\rangle \leq \epsilon.$$

## 2.4 The occupancy measure view

As mentioned, our algorithm `OMPO` will operate over the occupancy measure space defined as follows. Given a policy $\pi$, let us consider a trajectory $\{(S_h, A_h)\}_{h=1}^H$ generated as $S_1 \sim \nu_1$, $A_h \sim \pi_h(\cdot|S_h)$, $S_{h+1} \sim f(\cdot|S_h, A_h)$ for all $h \geq 1$. Then, the single player occupancy measure of $\pi$, denoted as $d_h^\pi \in \Delta_{\mathcal{S} \times \mathcal{A}}$, is defined at stage $h$ as $d_h^\pi(s, a) = \Pr(S_h = s, A_h = a)$.

We also define the occupancy measure conditioned on a particular initial state $d_{h|s_1}^\pi(s, a) = \Pr(S_h = s, A_h = a|S_1 = s_1)$. In addition, given the policies $\pi, \bar{\pi}$ and corresponding rollouts $\{(S_h, A_h)\}_{h=1}^H$ and $\{(S_h', A_h')\}_{h=1}^H$ from the same initial state $S_1 = S_1'$, the *joint* occupancy measure of $(\pi, \bar{\pi})$ at stage $h$ is defined as $d_h^{\pi,\bar{\pi}}(s, a, s', a') = \Pr(S_h = s, A_h = a, S_h' = s', A_h' = a')$.

The usefulness of the occupancy measures is that the expected value function at the initial state can be represented as an inner product between the reward function and the joint occupancy measure, i.e., $\langle \nu_1, V^{\pi,\bar\pi} \rangle = \sum_{h=1}^H \langle r, d_h^{\pi,\bar\pi} \rangle$. Moreover, given the structure of the game where the sequences of sentences and answers are generated independently by the two agents given an initial state $s_1 \in \mathcal{S}$, the joint occupancy measure at each step can be factorized as the product of the two agents occupancy measures given a particular $s_1$. In particular, we have $d_{h|s_1}^{\pi,\bar\pi}(s,a,s',a') = d_{h|s_1}^{\pi}(s,a) \cdot d_{h|s_1}^{\bar\pi}(s',a')$ for all $h,s,a,s',a'$. This makes possible to write the objective in a bilinear form, that is, $\langle \nu_1, V^{\pi,\bar\pi} \rangle = \mathbb{E}_{S_1 \sim \nu_1} \left[ \sum_{h,s,a,s',a'} d_{h|S_1}^{\pi}(s,a) r(s,a,s',a') d_{h|S_1}^{\bar\pi}(s',a') \right]$.

Moreover, we can characterize the set of the occupancy measures via $|\mathcal{S}|$ dimensional affine constraints. In particular, for each possible initial state $s_1$, the set

$$\mathcal{F}_{s_1} = \left\{ d = (d_1, \ldots, d_H) : \sum_a d_{h+1}(s,a) = \sum_{s',a'} f(s|s',a') d_h(s',a'), d_1(s) = \mathbb{1}\{s = s_1\} \right\}$$

describes the possible occupancy measures in the sense that for any element $d = (d_1, \ldots, d_H) \in \mathcal{F}_{s_1}$ there exists a policy $\pi \in \Pi$ such that $d_{h|s_1}^{\pi} = d_h$ for all $h \in [H]$. This is an elementary fact about MDP whose proof can be found in Puterman (1994). $\mathcal{F}$ is the product set of the Bellman flow constraints for a particular initial state, i.e. $\mathcal{F} = \times_{s_1 \in \mathrm{supp}(\nu_1)} \mathcal{F}_{s_1}$.

With this notation in place we can write the following program, which corresponds to Game lifted to the space of occupancy measures.

$$(d^\star, d^\star) = \arg\max_{d \in \mathcal{F}} \min_{d' \in \mathcal{F}} \mathbb{E}_{S_1 \sim \nu_1} \sum_{h=1}^H \sum_{s,a,s',a'} d_h(s,a|S_1) r(s,a,s',a') d_h'(s',a'|S_1). \tag{Occ-Game}$$

The policy pair $(\pi^\star, \pi^\star)$ solution of Game can be retrieved from the occupancy measure pair $(d^\star, d^\star)$ as $\pi^\star(a|s) = \frac{d^\star(s,a)}{\sum_a d^\star(s,a)}$. The advantage of the reformulation is that the program over occupancy measures is linear with affine constraints while Game is non convex non concave.

Moreover, lifting the problem to the occupancy measures turns out to be fundamentally important for enabling each agent to learn a policy conditioned only on their own state. This is different from the standard literature on Markov Games (Daskalakis et al., 2020; Wei et al., 2021; Alacaoglu et al., 2022), which assumes that both agents share a common state. Our idea, described in details in the next section, is to apply the optimistic algorithm from Joulani et al. (2017) to the reformulation of Game over occupancy measures. We present the resulting algorithm, i.e., `OMPO`, in Alg. 1.

## 3 Algorithm and convergence guarantees

Below we detail our algorithm, summarized in Sec. 3.1. The derivation is based on the optimistic online descent method applied on the reformulation of the optimization problem in the occupancy measures space. In particular, we will use that optimistic online mirror descent (Optimistic OMD) with one projection (Joulani et al., 2017). For a bilinear function $g : \mathcal{Z} \times \mathcal{W} \to \mathbb{R}$ such that $g(z,w) = \langle z, Aw \rangle$, optimistic OMD can be used to compute a saddle point $\min_{z \in \mathcal{Z}} \max_{w \in \mathcal{W}} g(z,w)$. In particular, the iterates for the $z$ player implemented with the Bregman divergence $\mathbb{D}$ induce by a Legendre potential with step size $\beta$ iterates as follows

$$z_{t+1} = \arg\min_{z \in \mathcal{Z}} \beta \langle 2Aw_t - Aw_{t-1}, z \rangle - \mathbb{D}(z, z_t).$$

The idea of optimism (Popov, 1980; Chiang et al., 2012; Rakhlin & Sridharan, 2013) has been used to obtain better regret bounds for slow changing loss sequences in online learning or to achieve minmax optimal rates in saddle point optimization. Our application falls into the latter category.

Specifically, we derive our algorithm given in Alg. 1 applying optimistic gradient descent ascent on the bilinear problem Occ-Game . The next section provides the convergence guarantees for our method.

---

**Algorithm 1** `OMPO` (Theory Version)

---

1: **input**: occupancy measure of reference policy $\pi^1$ denoted as $d^1$, preference oracle $\mathbb{P}$ (i.e. reward function $r$), learning rate $\beta$, Bregman divergence $\mathbb{D}$, iteration $T$
2: **for** $t = 1, 2, \ldots, T$ **do**
3:
$$d_h^{t+1} = \underset{d \in \mathcal{F}}{\arg\max} \, \beta \left\langle d, 2\mathbb{E}_{S', A' \sim d_h^t} r(\cdot, \cdot, S', A') - \mathbb{E}_{S', A' \sim d_h^{t-1}} r(\cdot, \cdot, S', A') \right\rangle - \mathbb{D}(d, d_h^t).$$

4: **end for**
5: $\pi_h^{\text{out}}(a|s) = \frac{\bar{d}_h(s,a)}{\sum_a \bar{d}_h(s,a)}$ with $\bar{d}_h = T^{-1} \sum_{t=1}^T d_h^t$ for all $h \in [H]$.
6: **Output :** $\pi^{\text{out}}$

---

### 3.1 Convergence guarantees of optimistic multi-step preference optimization (`OMPO`)

As the next theorem shows, in the ideal case where the updates can be computed exactly, Alg. 1 finds an $\epsilon$-approximate Nash equilibrium using fewer updates compared to a naive application of natural actor critic in this setting (see Alg. 3 in Appx. C) and to Swamy et al. (2024, Alg. 1). The proof can be found at Appx. D.3.

**Theorem 4** (Convergence of `OMPO`). *Consider Alg. 1 and let us assume that the occupancy measure of the reference policy $d^1$ is uniformly lower bounded by $\underline{d}$. Moreover, let $\mathbb{D}$ be $1/\lambda$ strongly convex, i.e. $\mathbb{D}(p||q) \geq \frac{\|p-q\|_1^2}{2\lambda}$. Then, by setting $T = \frac{10H \log \underline{d}^{-1}}{\beta \epsilon}$ and $\beta \leq \frac{1}{\sqrt{2\lambda}}$, we ensure that $(\pi^{\text{out}}, \pi^{\text{out}})$, i.e., the output of Alg. 1 is an $\epsilon$-approximate Nash equilibrium. Therefore, we need at most $\frac{10H \log \underline{d}^{-1}}{\beta \epsilon}$ policy updates.*

In addition, not only Swamy et al. (2024, Alg. 1) but also `OMPO` can be implemented using only one player since in a constant sum game, the max and min player produce the same iterates. The result is formalized as follows and the proof is deferred to Appx. D.4.

**Theorem 5.** *Consider a constant sum two-player Markov game with reward such that $r(s, a, s', a') = 1 - r(s', a', s, a)$, then for each $s_1 \in \text{supp}(\nu_1)$ the updates for $d$ in Alg. 1 coincides with the updates for the min player that uses the updates*

$$d_h^{t+1} = \underset{d \in \mathcal{F}}{\arg\max} \, \beta \left\langle d, 2\mathbb{E}_{S', A' \sim d_h^t} r(\cdot, \cdot, S', A') - \mathbb{E}_{S', A' \sim d_h^{t-1}} r(\cdot, \cdot, S', A') \right\rangle - \mathbb{D}(d, d_h^t).$$

It is important to notice that the above theorem uses the fact that shifting the reward by a constant does not change the optimal policy. This happens because all feasible occupancy measures have fixed total mass. For the first iteration, we initialize $d_h^0$ to be equal to $d_h^1$ for all $h$. Moreover, the next theorem shows that the last iterate converges asymptotically. The proof is deferred to Appx. D.5.

**Theorem 6.** *Assume that $\{d^t\}_{t=1}^\infty$ are the iterates generated by Alg. 1 with $\beta \leq 1/\sqrt{2\lambda}$ and that there exists a NE $d^\star$ such that $d^\star(s, a) > 0$. Then, their limit exists. Then $\{d^t\}_{t=1}^\infty$ converges to the set of Nash equilibria of Occ-Game .*

An important caveat of the above result is that it requires the existence of an equilibrium in the interior of the domain. In case the Nash equilibrium lies on the boundary our average iterate convergence guarantee still applies while the last iterate result becomes vacuous.

### 3.2 Efficient implementation

We can avoid the projection over the set $\mathcal{F}$ by implementing this update on the policy space (see Appendix E). We achieve such results following the techniques developed in Bas-Serrano et al. (2021); Viano et al. (2022) for specific choices of the Bregman divergence $\mathbb{D}$. In particular for the relative entropy, $\mathbb{D}(p, q) = \sum_{h=1}^H \sum_{s,a} p_h(s, a) \log \left( p_h(s,a)q_h(s) / q_h(s,a)p_h(s) \right)$ which is 1-strongly convex, we show in E.2 that the update in Alg. 1 can be implemented as follows

$$\pi_h^{t+1}(\cdot|s) \propto \pi_h^t(\cdot|s) \odot \exp \left( \beta_h Q_h^t(s, \cdot) \right), \quad Q_h^t(s, a) = \widetilde{r}^t(s, a) + \mathbb{E}_{s' \sim f(\cdot|s,a)} V_h^t(s'),$$

$$\widetilde{r}^t(s,a) = \sum_{s',a'} (2d_h^t(s',a') - d_h^{t-1}(s',a'))r(s,a,s',a'), \quad V_h^t(s) = \frac{1}{\beta_h} \log \sum_a \pi_h^t(a|s) \exp(\beta_h Q_h^t(s,a)),$$

where $\beta_h = \frac{\beta}{H-h+1}$. These updates for the value functions are known as soft-Bellman equation Ziebart (2010). The reward $\widetilde{r}^t$ has this particular form because of the particular optimistic mirror descent update that we are performing.

**Approximating the value function updates**   Unfortunately these updates suffer from numerical instabilities in practice but for $\beta \to 0$ we have that the regularized value functions $Q_h^t$ and $V_h^t$ tends to the standard state action and state value function respectively . Indeed as shown in the next theorem we have that $V_h^t(s) \to \langle \pi_h^t(a|s), Q_h^t(s,a) \rangle$ for $\beta \to 0$.

**Theorem 7.** *Let us denote $\beta_h = \frac{\beta}{H-h+1}$ and let us assume that the values $Q_h^t$ generated by the soft Bellman equations in Thm. 10 are uniformly upper bounded by $Q_{\max}$, and let us choose $\beta_h \leq \frac{1}{Q_{\max}}$ for all $h \in [H]$. Then, it holds that*

$$\langle \pi_h^t(\cdot|s), Q_h^t(s,\cdot) \rangle \leq \frac{1}{\beta_h} \log \sum_a \pi_h^t(a|s) \exp(\beta_h Q_h^t(s,a)) \leq \langle \pi_h^t(\cdot|s), Q_h^t(s,\cdot) \rangle + \beta_h Q_{\max}^2 .$$

Therefore, in practical implementation with small $\beta$ it is reasonable to approximate the regularized state action value functions with the standard single player state action value functions for the reward function $\widetilde{r}^t$ denoted with $Q_{h,\widetilde{r}}^\pi(s,a) = \mathbb{E}_\pi \left[ \sum_{\tau=h}^H \widetilde{r}^t(S_\tau, A_\tau)|S_h = s, A_h = a \right]$. Moreover, given the definition of $\widetilde{r}^t$, we can write $Q_{h,\widetilde{r}}^{\pi^t}$ as function of the joint action value functions as follows:

$$Q_{h,\widetilde{r}}^{\pi^t}(s,a) = 2\mathbb{E}_{S',A' \sim d_h^t} Q_h^{\pi^t,\pi^t}(s,a,S',A') - \mathbb{E}_{S',A' \sim d_h^{t-1}} Q_h^{\pi^t,\pi^{t-1}}(s,a,S',A').$$

In practice, the dynamics are unknown, so we use a standard Monte Carlo to approximate the state action value functions. For the first term, we sample $K$ pairs of trajectories from the same LLM ( with policy $\pi_h^t$) denoted $\left\{ (S_\tau^k, A_\tau^k) \right\}_{\tau=1,k=1}^{H,K}$ and $\left\{ (S_\tau'^{,k}, A_\tau'^{,k}) \right\}_{\tau=1,k=1}^{H,K}$ respectively. For the second term, we have to produce $K$ trajectories from the old policy $\pi^{t-1}$, let us denote this rollouts as $\left\{ (S_\tau^{\dagger,k}, A_\tau^{\dagger,k}) \right\}_{\tau=1,k=1}^{H,K}$ . At this point, we can produce the estimator whose unbiasedness is easy to be verified ( i.e. $\mathbb{E}\left[ \widehat{Q_h^t}(s,a) \right] = Q_{h,\widetilde{r}}^{\pi^t}(s,a)$).

$$\widehat{Q_h^t}(s,a) = \frac{1}{K} \sum_{k=1}^K \sum_{\tau=h}^H \left( 2\mathbb{P}([S_\tau^k, A_\tau^k] \succ [S_\tau'^{,k}, A_\tau'^{,k}]) - \mathbb{P}([S_\tau^k, A_\tau^k] \succ [S_\tau^{\dagger,k}, A_\tau^{\dagger,k}]) \right) \mathbb{1}_{\{S_1^k = s, A_1^k = a\}}. \quad (2)$$

At the initial, iteration, when $\pi^{t-1}$ is undefined we use the estimator $\widehat{Q_h^1}(s,a) = \frac{1}{K} \sum_{k=1}^K \sum_{\tau=1}^H \mathbb{P}([S_\tau^k, A_\tau^k] \succ [S_\tau'^{,k}, A_\tau'^{,k}]) \mathbb{1}_{\{S_1^k = s, A_\tau^k = a\}}.$

**Approximating the policy update**   The last obstacle for a practical implementation is the normalization constant in the policy update, which is intractable in practice. To circumvent this problem, we use the approach suggested in Wu et al. (2025), which treats the log of the unknown normalization constant as a tunable parameter.

The detailed pseudocode of our practical implementation is in Alg. 2. First, recall that our goal update with the estimated state action value function $\pi_h^{t+1}(\cdot|s) \propto \pi_h^t(\cdot|s) \odot \exp\left( \beta_h \widehat{Q_h^t}(s,\cdot) \right)$ could be implemented exactly as in the next equation if the state dependent normalization constant $Z_h^t(s)$ was computationally tractable, i.e., $\pi_h^{t+1}(a|s) = \frac{\pi_h^t(a|s) \exp\{\beta \widehat{Q_h^t}(s,a)\}}{Z_h^t(s)}$ .   This equation can be expressed equivalently as follows $\log \frac{\pi_h^{t+1}(a|s)}{\pi_h^t(a|s)} = \beta \widehat{Q_h^t}(s,a) - \log Z_h^t(s)$. Therefore, following Wu et al. (2025), we approximate the above equality with the following regression problem:

$$\pi^{t+1} = \arg\min_{\pi \in \Pi} \mathbb{E}_{\substack{S \sim \nu_1 \\ A \sim \pi(\cdot|S)}} \left[ \sum_{h=1}^H \left( \log \frac{\pi_h^{t+1}(A|S)}{\pi_h^t(A|S)} - \beta \widehat{Q_h^t}(S,A) + \log Z_h^t(S) \right)^2 \right].$$

Finally, to ensure computationally tractability we replace $\log Z_h^t(s)$ with $\beta \frac{H-h+1}{2}$ in all states $s$. Such heuristic is motivated by the following observation: If the preference between $a_h$ and $a_h'$ in Eq. (4) results in a tie, then with such $\log Z_h^t(s)$, the solution of Eq. (4) is $\pi^{t+1} = \pi^t$, leaving the model unchanged. In summary, we provide a practical version of `OMPO` in Alg. 2. For simplicity, we used a stationary policy whioch is a good approximation for large $H$ and we find to be sufficient to obtain convincing results.

---

**Algorithm 2** `OMPO` (Practical version)

---

**input**: reference policy $\pi^1$, preference oracle $\mathbb{P}$, learning rate $\beta$, number of generated samples $K$, horizon $H$, total iteration $T$, tunable bias term $\tau$.

**for** $t = 1, 2, \ldots, T$ **do**

    Sample $S_1^1 \sim \nu_1$.

    **for** $h = 1, 2, \ldots, H$ **do**

        Generate responses $A_h^1 \sim \pi^t(\cdot|S_h^1)$.

    **end for**

    Clear the dataset buffer $\mathcal{D}_t$.

    **for** $h = 1, 2, \ldots, H$ **do**

        Set $S_h^K = \cdots = S_h^2 = S_h^1$.

        Generate $K - 1$ conversations by sampling $A_{\hat{h}}^{2:K} \sim \pi^t(\cdot|S_{\hat{h}}^{2:K})$ for $\hat{h} \in [h, H]$.

        Estimate $\widehat{Q_h^t}$ via Eq. (2).

        Add $\{(S_h^1, A_h^k)\}_{k \in [K]}$ into $\mathcal{D}_t$.

    **end for**

    Update policy $\pi^{t+1} \leftarrow \arg\min_{\pi \in \Pi} \sum_{S, A \in \mathcal{D}_t} \left( \log \pi(A|S) - \log \pi^t(A|S) - \beta \widehat{Q}_1^t(S, A) + \beta \frac{H-h+1}{2} \right)^2$.

**end for**

**output**: $\pi^{T+1}$

---

## 4 Experiments

In this section, we provide several numerical results. Additional description of the dataset setup, detail on hardware setup, hyperparameter setup, evaluation setup, and ablation studies can be found in Appx. F Beyond comparing `OMPO` with recent algorithms from the literature we compare with a simpler multi step method based on actor critic dubbed `MPO`. We provide the derivation in Appx. C. This comparison serves to assess the importance of the formulation over occupancy measures and of the optimism in the policy update.

### 4.1 Tabular experiment

First, we consider a synthetic experiment in which the state action functions can be computed exactly for both `OMPO` and `MPO`. We generate 10 random gridworlds with a number of states and actions sample uniformly from the intervals $[1, 100]$ and $[2, 10]$. We plot the exploitability computed as $\max_\pi \left\langle \nu_1, V^{\pi, \pi^k} - V^{\pi^k \pi^k} \right\rangle$, which is a standard metric to evaluate the distance from a Nash equilibrium. In particular, when $(\pi^k, \pi^k)$ is a Nash equilibrium, the exploitability is 0. We can see that `OMPO` achieves very low exploitability after 100 updates while 2000 updates are needed by `MPO`. In this case, where the $Q$ functions can be computed exactly, we can appreciate the faster convergence rate of `OMPO` as described by Thm. 4.

### 4.2 Experiment on multi-turn conversation dataset

In this section, we test the proposed algorithms with multi-turn conversations in MT-bench-101 (Bai et al., 2024), see detailed description of this dataset at Appx. F.1. We choose Mistral-7B-Instruct-v0.2 as the base model (Jiang et al., 2023). We use a pre-trained PairRM [2] as the preference oracle. Specifically, given two conversations $[s_h, a_h]$ and $[s_h', a_h']$, PairRM will return a score that indicates the probability that $[s_h, a_h]$ is

---

[2] https://huggingface.co/llm-blender/PairRM

Table 2: Evaluation results on MT-bench-101 dataset. Mistral-7B-Instruct is selected as the base model. We can observe that both of the proposed algorithms MPO and OMPO considerably outperform the baseline in terms of the score (the higher the better). For OMPO, we omit the result for iteration 1, as it is the same as MPO. OMPO relies on information from two adjacent steps, and at the initial step there is no previous step, so it coincides with MPO.

| Model | Avg. | Perceptivity | | | | | Adaptability | | | | | | Interactivity | |
|---|---|---|---|---|---|---|---|---|---|---|---|---|---|---|
| | | CM | SI | AR | TS | CC | CR | FR | SC | SA | MR | GR | IC | PI |
| Base (Mistral-7B-Instruct) | 6.223 | 7.202 | 7.141 | 7.477 | 7.839 | 8.294 | 6.526 | 6.480 | 4.123 | 4.836 | 4.455 | 5.061 | 5.818 | 5.641 |
| DPO (iter=1) | 6.361 | 7.889 | 6.483 | 7.699 | 8.149 | 8.973 | 7.098 | 7.423 | 3.448 | **6.123** | 3.421 | 4.492 | 5.639 | 5.858 |
| DPO (iter=2) | 6.327 | 7.611 | 6.206 | 8.106 | 8.052 | 9.111 | 6.670 | 7.153 | 3.494 | 5.884 | 3.360 | 4.691 | 5.837 | 6.078 |
| DPO (iter=3) | 5.391 | 6.019 | 4.521 | 6.890 | 6.631 | 8.177 | 5.437 | 5.723 | 3.448 | 5.295 | 3.142 | 4.015 | 5.256 | 5.529 |
| SPPO (iter=1) | 6.475 | 7.432 | 7.464 | 7.714 | 8.353 | 8.580 | 6.917 | 6.714 | 4.136 | 5.055 | 4.403 | 5.400 | 6.036 | 5.966 |
| SPPO (iter=2) | 6.541 | 7.516 | 7.496 | 7.808 | 8.313 | 8.731 | 7.077 | 6.867 | 4.136 | 5.281 | 4.488 | 5.477 | 6.098 | 5.751 |
| SPPO (iter=3) | 6.577 | 7.575 | 7.547 | 7.944 | 8.365 | 8.797 | 7.040 | 6.865 | 4.442 | 5.185 | 4.346 | 5.394 | 6.092 | 5.906 |
| Step-DPO (iter=1) | 6.433 | 7.463 | 7.054 | 7.790 | 8.157 | 8.593 | 6.827 | 6.748 | 4.234 | 4.849 | 4.236 | 5.519 | 5.982 | 6.171 |
| Step-DPO (iter=2) | 6.553 | 7.616 | 7.043 | 7.925 | 8.147 | 8.662 | 6.790 | 6.878 | 4.331 | 5.048 | 4.366 | **5.734** | **6.391** | 6.254 |
| Step-DPO (iter=3) | 6.442 | 7.665 | 7.023 | 7.767 | 8.016 | 8.589 | 6.723 | 6.581 | 4.305 | 5.014 | 4.153 | 5.453 | 6.202 | **6.257** |
| MPO (iter=1) | 6.630 | 7.624 | **7.846** | 8.085 | 8.398 | 8.947 | 7.105 | 7.286 | 4.208 | 4.993 | 4.377 | 5.264 | 6.179 | 5.873 |
| MPO (iter=2) | 6.735 | 7.838 | 7.723 | 8.196 | **8.590** | 9.027 | 7.347 | 7.209 | 4.240 | 5.137 | 4.469 | 5.531 | 6.181 | 6.061 |
| MPO (iter=3) | 6.733 | **7.868** | 7.686 | **8.289** | 8.510 | 9.078 | 7.330 | 7.529 | **4.461** | 4.829 | 4.225 | 5.366 | 6.198 | 6.155 |
| OMPO(iter=2) | 6.736 | 7.733 | 7.723 | 8.257 | 8.478 | 9.122 | 7.300 | 7.421 | 4.123 | 5.288 | **4.506** | 5.513 | 6.179 | 5.923 |
| OMPO(iter=3) | **6.776** | 7.649 | 7.792 | 8.281 | 8.578 | **9.136** | **7.424** | **7.635** | 4.377 | 5.308 | 4.312 | 5.455 | 6.187 | 5.954 |

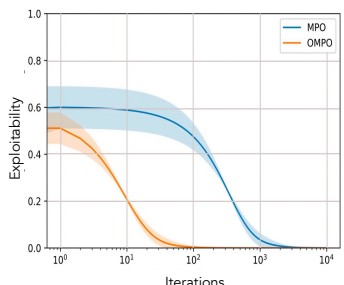

(a) Results in tabular experiments.

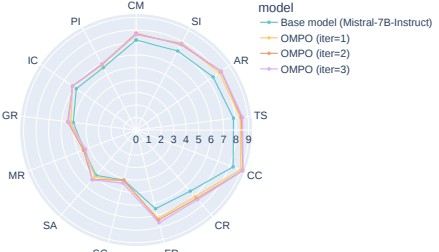

(b) Radar chart on different categories.

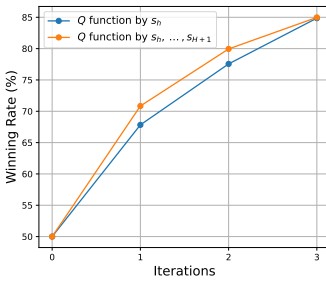

(c) Winning rate against the base model.

Figure 1: (a): Results in the tabular experiments. Curves are averages across 10 different randomly generated environments. The error bars report one standard deviation. (b): Result of OMPO on the MT-bench-101 dataset; (c) Winning rate against the base model with different approximations for the $Q$ functions. When optimizing $a_h$ at the $h$ step, only considering the preference of $s_h$ is sufficient compared to using $s_h, \ldots, s_{H+1}$.

better than $[s'_h, a'_h]$, which can be used to considered as the preference oracle $\mathbb{P}$ defined in the previous section. We select iterative DPO (Dong et al., 2024), iterative SPPO (Wu et al., 2025), and iterative Step-DPO as our baselines. For both iterative DPO and iterative SPPO, we sample $K = 5$ complete conversations starting from $s_1$, and estimate the winning rate $\mathbb{P}([s^k_{H+1}, a^k_{H+1}] \succ (s^{k'}_{H+1}, a^{k'}_{H+1})) \, \forall k, k' \in [K]$. Then we select both the best and worst conversations according to their winning rates against others, which is defined as $\frac{1}{K} \sum_{k'=1}^{K} \mathbb{P}([s^k_{H+1}, a^k_{H+1}] \succ [s^{k'}_{H+1}, a^{k'}_{H+1}])$ for the conversation $[s^k_{H+1}, a^k_{H+1}]$. Such a pair is used to train DPO while the winning rate is used to train SPPO. For both Step-DPO, MPO, and OMPO, we do the same strategy with starting at $s_h$. In MPO and OMPO, we estimate $Q(s_h, a_h, s_h, a'_h)$ by $\mathbb{P}([s_h, a_h] \succ [s_h, a'_h])$ to enhance the efficiency. For OMPO, the $Q^{\pi^t, \pi^{t-1}}$ term is estimated by calculating the winning rate between two answers (the best and the worst) generated by the current policy $\pi^t$ and the five answers previously generated by $\pi^{t-1}$. Each round of dia-

Table 3: Performance of math reasoning on MATH and GSM8K dataset across various models. `MPO` and `OMPO` achieve performance comparable to Step-DPO (Lai et al., 2024) without requiring the ground truth label of the dataset during fine-tuning while Lai et al. (2024) requires. Additionally, `MPO` and `OMPO` only need access to a Llama-3-based reward model (RM) to compare two answers whereas Step-DPO Lai et al. (2024) requires GPT-4 to locate and identify the incorrect reasoning step in an answer, which is a considerably more difficult task than comparison.

| Method | Additional info on incorrect step | Auxiliary Autoregressive Language Model | Average | GSM8K | Math |
|---|---|---|---|---|---|
| Base (Qwen2-7B-Instruct) | - | - | 0.7049 | 0.8559 | 0.5538 |
| Step-DPO (Lai et al., 2024) | ✓ | ✓ (Require GPT-4) | 0.7258 | 0.8680 | **0.5836** |
| Step-DPO | ✗ | ✗ (Require Llama-3 RM) | 0.7184 | 0.8749 | 0.5618 |
| MPO | ✗ | ✗ (Require Llama-3 RM) | 0.7260 | 0.8734 | 0.5786 |
| OMPO | ✗ | ✗ (Require Llama-3 RM) | **0.7283** | **0.8779** | 0.5786 |

logue is rated on a scale of 1 to 10 by GPT-4o mini, with the mean score reported for each dialogue. All methods are run for a total of 3 iterations. The results are summarized in Tab. 2, showing significant improvements over the baselines with the proposed `MPO` and `OMPO` approaches. In Fig. 1(b), we present the Radar chart on different categories and we can see that the proposed `OMPO` leads to improvements generally along the iterations. Fig. 1(c) shows that using the entire trajectory to estimate the $Q$ function can lead to subtle improvement at the first two iterations while it finally achieves a similar winning rate when compared to the one that only use one step.

### 4.3 Experiment on math reasoning dataset

As discussed in Sec. 2, our framework can also cover the alignment of chain-of-thought reasoning. In this section, we validate the proposed methods in two widely used math reasoning datasets: MATH Hendrycks et al. (2021) and GSM8K Cobbe et al. (2021). We use Qwen2-7B-Instruct as the base model and follow the same evaluation procedure as in Lai et al. (2024). We adopt the dataset for alignment from Lai et al. (2024), which contains 10795 samples of augmented mathematical problems from MetaMath (Yu et al., 2024) and MMIQC (Liu et al., 2024b)[3]. For both `MPO` and `OMPO`, we select the Llama-3-based model[4] as the preference oracle. For Step-DPO, we implement two versions. The first version is using the Llama-3-based model as the preference oracle and follows the same procedure as `MPO` and `OMPO`. The second version is using the checkpoint provided in Lai et al. (2024). The result is provided in Tab. 3, showing that the proposed methods achieve performance comparable to Step-DPO (Lai et al., 2024).

## 5 Conclusion

This work presents a novel framework to enhance the preference alignment of LLMs in multi-step setting by casting the alignment process as a two-player Markov game. In particular, we provided a new formulation of the problem on the occupancy measure space and propose an optimistic mirror descent ascent scheme to solve it. A natural open direction is to investigate the rate for the last iterate of Alg. 1 for which our work only establishes an asymptotic convergence result. Moreover, we think it would be interesting to understand if our techniques still applies to the infinite horizon discount setting and to general sum matrices. A second interesting direction is to find other applications of our formulation of learning from general preferences over the space of occupancy measures. In addition, at theoretical level, it is interesting to investigate whether the same conclusion offered by our work can extend to the infinite horizon setting. The main obstacle in that direction is to establish the analogous result to the factorization of the occupancy measure, which we used to derive the game formulation given in Eq. (Occ-Game ). For this reason, we think that the analysis of the infinite horizon setting will require new conceptual tools to be carried out. From the practical point of view, future work could extend our method to vision-language models (VLMs) for aligning both text and image

---

[3]https://huggingface.co/datasets/xinlai/Math-Step-DPO-10K
[4]https://huggingface.co/RLHFlow/pair-preference-model-LLaMA3-8B

modalities. One can also apply our approach in the AI safety domain, particularly as a potential multi-step defense mechanism against jailbreak attacks.

## 6 Broader impact

In this work, we propose novel algorithms for multi-step alignment in LLMs and establish their theoretical guarantees. Our framework can make LLMs better at understanding and following complex instructions over time. Our method could help improve AI systems used in education, math reasoning, finance reasoning, customer service, or other areas where multi-step inference matter. We do not create any new benchmarks for human preferences nor solicit human preferences for this study. As such, we do not expect any potential violations of ethical standards, including those concerning the use of human data. Our contributions are primarily methodological and theoretical analysis of the convergence, and we have taken care to ensure that our work complies with all relevant ethical guidelines.

## Acknowledgements

This work was supported by Hasler Foundation Program: Hasler Responsible AI (project number 21043). This work was supported by the Swiss National Science Foundation (SNSF) under grant number 200021_205011. This work is funded (in part) through a PhD fellowship of the Swiss Data Science Center, a joint venture between EPFL and ETH Zurich. This research was sponsored by the Army Research Office and was accomplished under Grant Number W911NF-24-1-0048. , Gu is partially supported by the National Science Foundation IIS-2008981, CPS-2312094, DMS-2323113, IIS-2403400 and the Sloan Research Fellowship. Part of this work was done during Yongtao Wu and Zhenyu Zhu's visit at UCLA.

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

## Contents of the Appendix

The Appendix is organized as follows:

- In Appx. A, we summarize the symbols and notation used in this paper.

- In Appx. B we provide a complete overview of the related works, elaborate on the difference and contribution of our work, and provide preliminaries on single-step RLHF.

- We describe `MPO` with natural actor-critic in Appx. C.

- In Appx. D, we provide the proofs for the theoretical results.

- Appx. E shows the implementation of Algorithm 1 with updates over policies.

- Appx. F provides supplementary material on the numerical experiments.

- We provide more discussion on Eq. (Game) in Appx. G.

Table 4: Core symbols and notations used in this paper.

| Symbol | Dimension(s) & range | Definition |
|---|---|---|
| $x_h$ | - | Prompt at step $h$ |
| $a_h$ | - | Specific Answer (action) at step $h$ |
| $A_h$ | - | An answer (action) sample from a certain distribution at step $h$ |
| $s_h$ | - | Specific state at step $h$ |
| $S_h$ | - | A state sampled from a certain distribution at step $h$ |
| $s_1(s_h)$ | - | The only initial state that can lead to $s_h$ |
| $\pi$ | | Language model (policy) |
| $\nu_1$ | | Initial distribution of state $s_1$ |
| $d_h^\pi(s, a)$ | $[0, 1]$ | Occupancy measure of $\pi$ at stage $h$ |
| $f$ | | Transition function |
| $\Pr(s_h = s, a_h = a)$ | $[0, 1]$ | Joint probability of $s_h = a$ and $a_h = a$ |
| $o$ | $\{0, 1\}$ | Preference oracle |
| $\mathbb{P}([s, a] \succ [s', a'])$ | $[0, 1]$ | Winning probability of $[s, a]$ against $[s', a')]$ |
| $D(p, q)$ | | KL divergence of two probability distributions $p$ and $q$ |
| $\mathbb{D}(p, q)$ | | Bregman Divergences between two points $q$ and $p$ |
| $\mathcal{D}_t$ | | Dataset buffet at iteration $t$ |
| $\Delta_{\mathcal{X}}$ | $[0, 1]^{|\mathcal{X}|}$ | Set of probability distributions over the set $\mathcal{X}$ |
| $\odot$ | - | Hadamard product between two vectors |
| $\mathcal{O}$, $o$, $\Omega$ and $\Theta$ | - | Standard Bachmann–Landau order notation |

We additionally use a compact notation for representing the Bellman flow constraints. We denote by $E \in \mathbb{R}^{|\mathcal{S}| \times |\mathcal{A}||\mathcal{S}|}$ the matrix such that $(Ez)(s, a) = z(s)$ for all vectors $z \in \mathbb{R}^{|\mathcal{S}|}$. Additionally, we denote by $F$ the matrix such that $(Fz)(s, a) = \sum_{s'} f(s'|s, a)z(s')$ for all vectors $z \in \mathbb{R}^{|\mathcal{S}|}$.

## A  Symbols and notation

We include the core symbols and notation in Tab. 4 to facilitate the understanding of our work.

## B  Related work

In this section, we present an overview of the related literature, discussion on the differences with related literatures, and preliminary on single-step RLHF.

### B.1 Overview of related work

**RLHF under Bradley-Terry model.** Over the years, significant strides have been made towards developing RLHF algorithms from various perspectives under the Bradley-Terry (BT) model Bradley & Terry (1952). Earlier RLHF pipelines usually included supervised fine-tuning, learning a reward model, and reinforcement learning optimization with PPO (Ziegler et al., 2019; Stiennon et al., 2020; Bai et al., 2022; Ouyang et al., 2022). Due to the instability and scaling issues of such a pipeline, direct alignment methods such as DPO have been proposed to bypass the training of the reward model (Rafailov et al., 2023). Several follow-up methods, such as generalized preference optimization (Tang et al., 2024), use offline preference data to directly optimize pairwise preferences against a fixed opponent. A number of works have proposed reference-model-free method (Meng et al., 2024; Hong et al., 2024). In Meng et al. (2024), the impact of sequence length is mitigated by averaging the likelihood over the length of the sequence. In the multi-step scenario, several multi-step variants of DPO are introduced in the math reasoning task. Lu et al. (2024) initiate from an intermediate step in a correct reasoning process and increase the temperature to produce a flawed reasoning path leading to an incorrect answer. Meanwhile, Lai et al. (2024) leverage GPT-4 to detect the first incorrect step in a multi-step reasoning trajectory, then regenerate from that point to obtain the correct path. Together, these serve as the pair of samples for DPO.

**RLHF under general preferences.** The reward model in the BT model inherently implies transitivity in preferences. However, human preferences, especially the resulting averaged human preferences from populations, are usually nontransitive (Tversky, 1969; Gardner, 1970). To this end, Azar et al. (2024) outline a general framework for RLHF starting from general preference optimization and shows that DPO is a special case with the assumption of BT model. They further proposed IPO without such an assumption. Subsequently, Munos et al. (2024) try to solve the alignment of non-transitive general preferences using two-player Nash learning in a bandit setting. In their work, preferences are regularized through KL divergence to a reference policy, and they prove the convergence of the last iterative. In Swamy et al. (2024), multi-step alignment is considered while preference signals are only applied at the final step. Swamy et al. (2024) do not demonstrate the effectiveness of this framework in large language models. Wu et al. (2025) propose SPPO, studying bandit alignment under general preferences. They introduce a novel loss function that increases the log-likelihood of the selected response while decreasing that of the rejected response, in contrast to DPO. Rosset et al. (2024) start with the Nash learning framework and propose Online DPO, which is an iterative version of DPO. Wang et al. (2023) provide theoretical analysis on multi-step RLHF under general preference while practice application is not explored. In Wang et al. (2023), the preference signal is given for the entire trajectory of an MDP while in this paper it is step-wise. Shani et al. (2024) study multi-step alignment under general preferences. However, unlike their approach where only preferences at the final states are considered, our work is built on a two-player Markov game which assumes that human preference is received at each step. Additionally, we leverage the optimistic online gradient descent to achieve a better convergence rate than Wang et al. (2023); Shani et al. (2024), and utilize Monte Carlo estimation with a small-scale pairwise reward model, avoiding the need for an additional function approximator for the critic network. Our contribution is compared to the recent literature on the same topic in Tab. 1.

**Two-player markov game & optimistic online gradient descent.** Two-player Markov games have been widely studied since the seminal work (Shapley, 1953). Particularly relevant to our work is the research line on policy gradient algorithms for two-player Markov games such as Daskalakis et al. (2020); Wei et al. (2021); Alacaoglu et al. (2022). Our `OMPO` is strictly related to the idea of optimistic online gradient descent (Popov, 1980; Chiang et al., 2012; Rakhlin & Sridharan, 2013) originally proposed in online learning to achieve small regret in case of slow varying loss sequences. Our update that uses only one projection per update was proposed in Joulani et al. (2017). The name of our method is due to a similar algorithm introduced in the context of variational inequalities by Malitsky & Tam (2020).

**Token-level preference optimization.** A line of work formulates the alignment of contextual bandit problems in LLMs (Example.1) from token-level MDPs perspective (Rafailov et al., 2024; Zeng et al., 2024; Liu et al., 2024a). In Rafailov et al. (2024), by defining the reward at each token before the terminal token as the generation likelihood and using the maximum entropy RL objective, the authors derive the original objective of DPO from a new perspective that incorporates token-level rewards. Zeng et al. (2024) assume that the reward for a response can be decomposed into token-level rewards at each token. Then they

design a token-level objective function based on Trust Region Policy Optimization, adding token-level KL divergence constraints to the DPO objective in the final algorithm. More recently, Liu et al. (2024a) study how the difference in average rewards between chosen and rejected responses affects the optimization stability, designing a new algorithm where importance sampling weights are assigned to each token-level reward. There are two main differences between the multi-step alignment approach in our work and those in previous work. First, while Rafailov et al. (2024); Zeng et al. (2024); Liu et al. (2024a) develop alignment methods based on the Bradley-Terry model with transitive rewards, our framework is motivated by a two-player game with relative rewards. Secondly, although Rafailov et al. (2024); Zeng et al. (2024); Liu et al. (2024a) formulate the alignment process as an MDP, their final objective is tailored to a contextual bandit problem in LLMs. In contrast, our objective is designed for a multi-step alignment problem, suited for multi-turn conversation or chain-of-thought reasoning.

## B.2 Discussion on the difference from SPPO

Next, we elaborate on the difference with SPPO (Wu et al., 2025) below: Firstly, the theoretical analysis of the proposed MPO differs from that of SPPO due to differences in the settings. SPPO considers the contextual bandit problem and builds its analysis based on the game matrix from Freund & Schapire (1999). In our case, however, we frame the problem as a Markov game and employ a distinct theoretical analysis apart from Freund & Schapire (1999). Specifically, in our proof, we (i) use the performance difference lemma to rewrite the global regret as weighted average of local regrets and (ii) control the local regrets with multiplicative weights updates. Secondly, a new algorithm, OMPO, is developed in this work with a novel theoretical guarantee. In the case where the horizon $H = 1$, the update of OMPO reduces to

$$\pi^{t+1}(a|s) \propto \pi^t(a|s) \exp\left[\beta(2\mathbb{P}(a \succ \pi^t(\cdot|s)) - \mathbb{P}(a \succ \pi^{t-1}(\cdot|s)))\right],$$

while the update of SPPO is

$$\pi^{t+1}(a|s) \propto \pi^t(a|s) \exp\left[\beta(\mathbb{P}(a \succ \pi^t(\cdot|s)))\right].$$

As a result, OMPO enables $\mathcal{O}(\epsilon^{-1})$ policy updates to converge to an $\epsilon$-approximate Nash equilibrium instead of $\mathcal{O}(\epsilon^{-2})$, according to our theoretical analysis.

## B.3 Preliminary on single-step RLHF

In this section, we review the earlier methods in single-step RLHF. Classical RLHF methods (Ziegler et al., 2019; Ouyang et al., 2022) assume that the preference oracle can be expressed by an underlying Bradley-Terry (BT) reward model (Bradley & Terry, 1952), i.e.,

$$\mathbb{P}([x_1, a_1] \succ [x_1, a_1']) = \sigma(r(x_1, a_1) - r(x_1, a_1')).$$

Thus, one can first learn a reward model and optimize the policy based on the following KL-constrained RL objective with PPO:

$$\pi^\star = \arg\max_\pi \mathbb{E}_{X_1 \sim \nu_1, A_1 \sim \pi(\cdot|X_1)}(r(X_1, A_1) - \beta D(\pi(\cdot|X_1), \pi_{\text{ref}}(\cdot|X_1))),$$

where $\beta$ is a parameter controlling the deviation from the reference model $\pi_{\text{ref}}$. Another line of work, e.g., DPO (Rafailov et al., 2023), avoids explicit reward modeling and optimizes the following objective over pair-wise preference data $(X_1, A_1^w, A_1^l)$.

$$\pi^\star = \arg\max_\pi \mathbb{E}_{(X_1, A_1^w, A_1^l) \sim \mathcal{D}}\left[\log \sigma\left(\beta \log \frac{\pi(A_1^w|X_1)}{\pi_{\text{ref}}(A_1^w|X_1)} - \beta \log \frac{\pi(A_1^l|X_1)}{\pi_{\text{ref}}(A_1^l|X_1)}\right)\right].$$

More recently, several studies (Swamy et al., 2024; Munos et al., 2024; Wu et al., 2025; Zhang et al., 2024; Rosset et al., 2024) have circumvented the Bradley-Terry (BT) assumption by directly modeling the general oracle $\mathbb{P}$, avoiding the reliance on the reward model which is transitive. Specifically, the goal is to identify the Nash equilibrium (or von Neumann winner) of the following two-player constant-sum game:

$$(\pi^*, \pi^*) = \arg\max_\pi \min_{\pi'} \mathbb{E}_{X_1 \sim \nu_1, A_1 \sim \pi(\cdot|X_1), A_1' \sim \pi'(\cdot|X_1)} \mathbb{P}([X_1, A_1] \succ [X_1, A_1']).$$

---

**Algorithm 3** `MPO` (Theoretical Version)

---

1: **input**: reference policy $\pi^1$, preference oracle $\mathbb{P}$, learning rate $\beta = \sqrt{\frac{\log \pi^{-1}}{TH^2}}$, total iteration $T$

2: **for** $t = 1, 2, \ldots, T$ **do**

3:
$$\pi_h^{t+1}(a|s) \propto \pi_h^t(a|s) \exp \left[ \beta \mathbb{E}_{S', A' \sim d_h^{\pi^t}|s_1(s)} Q_h^{\pi^t, \pi^t}(s, a, S', A') \right]$$

4: **end for**

5: **output**: $\bar{\pi}^T$ (s.t. $d_h^{\bar{\pi}^T} = \frac{1}{T} \sum_{t=1}^T d_h^{\pi^t}, \ \forall h \in [H].$).

---

## C  `MPO` with natural actor-critic

This section presents our first method to find an approximate solution to Game. In order to find an $\epsilon$-approximate Nash equilibrium, the MPO method builds upon the next lemma which decomposes the difference of two value functions to the $Q$ function at each step. Lemma 1 is the extension of Kakade & Langford (2002) to the multi-agent setting where the dynamics are controlled independently by each player but the reward depends on the joint-state action tuple. In Kakade & Langford (2002), the $Q$ function is a function of only one state-action pair while in our setting the $Q$ function is based on two state-action pairs.

**Lemma 1** (Value difference lemma (Adapted from Kakade & Langford (2002))). *For a finite horizon MDP with initial distribution $\nu_1$ it holds that:*

$$\left\langle \nu_1, V^{\pi, \bar{\pi}} - V^{\pi', \bar{\pi}} \right\rangle = \mathbb{E}_{S_1 \sim \nu_1} \sum_{h=1}^H \mathbb{E}_{S \sim d_h^{\pi}|S_1} \left[ \left\langle \mathbb{E}_{S', A' \sim d_h^{\bar{\pi}}|S_1} Q_h^{\pi', \bar{\pi}}(S, \cdot, S', A'), \pi_h(\cdot|S, S_1) - \pi_h'(\cdot|S, S_1) \right\rangle \right].$$

The proof can be found at Appx. D.1. In our setting, the initial state $S_1$ is a deterministic function of the state $S$ so we can remove $S_1$ from the conditioning in the policy[5]. To highlight this fact we denote, for all $s \in \mathcal{S}$ as $s_1(s)$ the only initial state that can lead to $s$ . By setting $\pi' = \bar{\pi} = \pi^t$ in Lemma 1 and $\pi = \pi^\star$ and summing from $t = 1$ to $T$ we obtain:

$$\sum_{t=1}^T \left\langle \nu_1, V^{\pi^\star, \pi^t} - V^{\pi^t, \pi^t} \right\rangle = \mathbb{E}_{s_1 \sim \nu_1} \sum_{h=1}^H \sum_{t=1}^T \mathbb{E}_{s \sim d_h^{\pi^\star}|s_1}$$

$$\left[ \left\langle \mathbb{E}_{s', a' \sim d_h^{\pi^t}|s_1} Q_h^{\pi^t, \pi^t}(s, \cdot, s', a'), \pi_h^\star(\cdot|s) - \pi_h^t(\cdot|s) \right\rangle \right].$$

Since the sum over $t$ commutes with the expectation, we see that we can decompose the global regret $\sum_{t=1}^T \left\langle \nu_1, V^{\pi^\star, \pi^t} - V^{\pi^t, \pi^t} \right\rangle$ into a weighted sum of local regrets at each stage $h \in [H]$. Therefore, we can control the global regret implementing at each state online mirror descent updates (Warmuth et al. 1997, Orabona 2023, Chapter 6, Cesa-Bianchi & Lugosi 2006), i.e., implementing the following update:

$$\pi_h^{t+1}(\cdot|s) = \arg\max_\pi \langle \pi(\cdot|s), \mathbb{E}_{S', A' \sim d_h^{\pi^t}|s_1(s)} Q_h^{\pi^t, \pi^t}(s, \cdot, S', A') \rangle - \beta D(\pi(\cdot|s), \pi_h^t(\cdot|s)),$$

where $\beta$ is a learning rate. The solution has the following form: $\pi_h^{t+1}(a|s) \propto \pi_h^t(a|s) \exp\{\beta \mathbb{E}_{S', A' \sim d_h^{\pi^t}|s_1(s)} Q_h^{\pi^t, \pi^t}(s, a, S', A')\}$, which corresponds to natural actor-critic (Peters & Schaal, 2008) that utilizes a softmax-based method for updating policies. The number of policy updates needed by the ideal version of `MPO` (see Alg. 3) can be bounded as follows and the proof can be found at Appx. D.2.

---

[5]This is motivated by practical LLM training, where system prompts such as "user" and "assistant" are inserted before every $x_h$ and $a_h$, respectively. As a result, one can infer a unique $s_1$ for every $s$. The conditioning of the policy on the initial state might appear unusual at the first glance but it is in fact common in the setting of Contextual MDPs (see for example Levy et al. (2023)). Indeed, the initial state $s_1$ could be interpreted as a context and we optimize over policies that depend on both the initial context and the current state.

---

**Algorithm 4** `MPO` (Practical version)

---

1: **input**: reference policy $\pi^1$, preference oracle $\mathbb{P}$, learning rate $\beta$, number of generated samples $K$, horizon $H$, total iteration $T$.
2: **for** $t = 1, 2, \ldots, T$ **do**
3:     Sample $s_1^1 \sim \nu_1$.
4:     **for** $h = 1, 2, \ldots, H$ **do**
5:         Generate responses $A_h^1 \sim \pi^t(\cdot|S_h^1)$.
6:     **end for**
7:     Clear the dataset buffer $\mathcal{D}_t$.
8:     **for** $h = 1, 2, \ldots, H$ **do**
9:         Set $S_h^K =, \ldots, = S_h^2 = S_h^1$.
10:         Generate $K - 1$ conversations by sampling $A_{\hat{h}}^{2:K} \sim \pi^t(\cdot|S_{\hat{h}}^{2:K})$ for $\hat{h} \in [h, H]$.
11:         Estimate $\mathbb{E}_{A_h^{k'}} Q^{\pi^t, \pi^t}(S_h^1, A_h^k, S_h^1, A_h^{k'}), \forall k, k' \in [K]$ via Eq. (4) with query to $\mathbb{P}$.
12:         Fill out $\mathcal{D}_t$ with the following data pair $\left\{ (S_h^1, A_h^k, \mathbb{E}_{A_h^{k'}} Q^{\pi^t, \pi^t}(S_h^1, A_h^k, S_h^1, A_h^{k'}) ) \right\}_{k \in [K]}$,
13:     **end for**
14:     Optimize $\pi_{t+1}$ over $\mathcal{D}_t$ according to $\pi^{t+1} \leftarrow \arg\min_\pi \mathbb{E}\left( \log\left( \frac{\pi(A_h^k|S_h^1)}{\pi^t(A_h^k|S_h^1)} \right) - \beta\left( \mathbb{E}_{A_h^{k'}} Q^{\pi^t, \pi^t}(S_h^1, A_h^k, S_h^1, A_h^{k'}) - \frac{H-h+1}{2} \right) \right)^2$.
15: **end for**
16: **output**: $\pi^{T+1}$

---

**Theorem 8.** *Consider Alg. 3 and assume that the reference policy is uniformly lower bounded by $\underline{\pi}$, then there exists a policy $\bar{\pi}^T$ such that $d_h^{\bar{\pi}^T} = \frac{1}{T} \sum_{t=1}^T d_h^{\pi^t}, \forall h \in [H]$, and it holds that for $T = \frac{16 H^4 \log \underline{\pi}^{-1}}{\epsilon^2}$ the policy pair $(\bar{\pi}^T, \bar{\pi}^T)$ is an $\epsilon$-approximate Nash equilibrium. Therefore, Alg. 3 outputs an $\epsilon$-approximate Nash equilibrium after $\frac{16 H^4 \log \underline{\pi}^{-1}}{\epsilon^2}$ policy updates.*

**Remark 2.** *The above result generalizes the $\mathcal{O}(H^2 \epsilon^{-2})$ bound on the policy updates proven in Swamy et al. (2024) in the setting of terminal-only reward. The additional $H^2$ factor in our theorem is due to considering rewards that are not terminal-only. In Thm. 4 we show that Alg. 1 improves the number of policy updates needed to converge to an $\epsilon$-approximate Nash equilibrium to $\mathcal{O}(H \epsilon^{-1})$.*

**Practical relaxations.** For the above theorem, `MPO` requires the access of the $Q$ function, which is unknown. Next, we are going to develop a practical algorithm to efficiently estimate the $Q$ function and implement Alg. 3. Equivalently, the update in Alg. 3 can be written as

$$\pi_h^{t+1}(a|s) = \frac{\pi_h^t(a|s) \exp\{\beta \mathbb{E}_{S', A' \sim d_h^{\pi^t}|s_1(S)} Q_h^{\pi^t, \pi^t}(s, a, S', A')\}}{Z_h^t(s)}, \tag{3}$$

where $Z_h^t(S)$ is the partition function. Next, we express Eq. (3) as follows for all $s, a \in \mathcal{S} \times \mathcal{A}$:

$$\log \frac{\pi_h^{t+1}(a|s)}{\pi_h^t(a|s)} = \beta \mathbb{E}_{S', A' \sim d_h^{\pi^t}|s_1(s)} Q_h^{\pi^t, \pi^t}(s, a, S', A') - \log Z_h^t(s).$$

Next, following Wu et al. (2025), we approximate the equation above with an approximate solution of the following optimization program:

$$\pi^{t+1} = \arg\min_\pi \sum_{h=1}^H \mathbb{E}_{\substack{S_1 \sim \nu_1 \\ (S_h, A_h) \sim d_h^{\pi^t}|S_1}} \left[ \log \frac{\pi(A_h|S_h)}{\pi_h^t(A_h|S_h)} - (\mathbb{E}_{S', A' \sim d_h^{\pi^t}|S_1} Q_h^{\pi^t, \pi^t}(S_h, A_h, S', A') - \log Z_h^t(S_h)) \right]^2.$$

Unfortunately, solving the above minimization exactly is out of hope. The first difficulty is the efficient estimation of $\mathbb{E}_{S', A' \sim d_h^{\pi^t}|s_1} Q_h^{\pi^t, \pi^t}(S_h, A_h, S', A')$. In particular, since $S'$ and $S$ are sampled from the same

distribution, we will sample $A'$ from the state $S_h$ and use the Monte Carlo estimator:

$$
\begin{aligned}
&\mathbb{E}_{A' \sim \pi^t(\cdot|S_h)} Q_h^{\pi^t, \pi^t}(S_h, A_h, S_h, A') \\
&\approx \frac{1}{K} \sum_{k=1}^{K} \sum_{\hat{h}=h}^{H} \mathbb{P}([S_{\hat{h},k}, A_{\hat{h},k}] \succ [S'_{\hat{h},k}, A'_{\hat{h},k}]) \mathbb{1}_{\left\{S_{h,k}=S'_{h,k}=S_h, A_{h,k}=A_h\right\}},
\end{aligned}
\tag{4}
$$

where the sequences $\left\{(S_{\hat{h},k}, A_{\hat{h},k}, S'_{\hat{h},k}, A'_{\hat{h},k})\right\}_{\hat{h}=h}^{H}$ for $k \in [K]$ are generated by rollouts of the policies pair $(\pi^t, \pi^t)$. The second difficulty is $Z_h^t(s)$, which is difficult to compute for large action spaces. In all states $s$, we replace $\log Z_h^t(s)$ with $\beta \frac{H-h+1}{2}$. Such heuristic is motivated by the following observation: If the preference between $a_h$ and $a'_h$ in Eq. (4) results in a tie, then with such $\log Z_h^t(s)$, the solution of Eq. (4) is $\pi^{t+1} = \pi^t$, leaving the model unchanged. In summary, we provide a practical version of `MPO` in Alg. 4. In practice, we used a stationary policy that we find to be sufficient to obtain convincing results.

## D    Proofs

**Lemma 2.** *(Adapted from Puterman (1994)) The pair-wise value function and pair-wise Q-value function satisfy the Bellman equation, i.e., for all $h \in [H]$: $Q_h^{\pi,\pi'}(s,a,s',a') = r(s,a,s',a') + \mathbb{E}_{\hat{S} \sim f(\cdot|s,a), \bar{S} \sim f(\cdot|s',a')}[V_{h+1}^{\pi,\pi'}(\hat{S}, \bar{S})]$ and $V_h^{\pi,\pi'}(s,s') = \mathbb{E}_{A \sim \pi_h(\cdot|S), A' \sim \pi'_h(\cdot|S')} Q_h^{\pi,\pi'}(s,a,A',A')$.*

*Proof.* By the definition of the state action value function for the policy pair $(\pi, \pi')$ we have that

$$
Q_h^{\pi,\pi'}(s,a,s',a') = r(s,a,s',a') + \mathbb{E}\Big[\sum_{h'=h+1}^{H} r(S_{h'}, A_{h'}, S'_{h'}, A'_{h'})\Big].
$$

Now, using tower property of the expectation we have that

$$
\begin{aligned}
&Q_h^{\pi,\pi'}(s,a,s',a') \\
&= r(s,a,s',a') + \mathbb{E}_{S'' \sim f(\cdot|s,a), \bar{S} \sim f(\cdot|s',a')}\Big[\mathbb{E}\Big[\sum_{h'=h+1}^{H} r(S_{h'}, A_{h'}, S'_{h'}, A'_{h'})|S_{h+1}=S'', S'_{h+1}=\bar{S}\Big]\Big] \\
&= r(s,a,s',a') + \mathbb{E}_{S'' \sim f(\cdot|s,a), \bar{S} \sim f(\cdot|s',a')}\Big[V^{\pi,\pi'}(S'', \bar{S})\Big],
\end{aligned}
$$

where the last equality follows from the definition of the state value function. $\square$

### D.1    Proof of Lemma 1

*Proof.* Let us consider the Bellman equation in vectorial form for the policy pair $(\pi', \bar{\pi})$, that is

$$
r_h + F V_{h+1}^{\pi',\bar{\pi}} = Q_h^{\pi',\bar{\pi}},
$$

where $F$ denoted the transition matrix induced by the transition function $f : \mathcal{S}^2 \times \mathcal{A} \to \Delta_{\mathcal{S} \times \mathcal{S}}$. Now, multiplying by the occupancy measure of the policy pair $(\pi, \bar{\pi})$ at stage $h$ we obtain

$$
\left\langle d_h^{\pi,\bar{\pi}}, r_h \right\rangle + \left\langle d_h^{\pi,\bar{\pi}}, F V_{h+1}^{\pi',\bar{\pi}} \right\rangle = \left\langle d_h^{\pi,\bar{\pi}}, Q_h^{\pi',\bar{\pi}} \right\rangle.
$$

At this point, using the Bellman flow constraints Puterman (1994), it holds that

$$
F^T d_h^{\pi,\bar{\pi}} = E^T d_{h+1}^{\pi,\bar{\pi}},
$$

where $E \in \mathbb{R}^{|\mathcal{S}|^2|\mathcal{A}| \times |\mathcal{S}|^2}$ such that $(E^T V)(s,a) = V(s)$ for all $V \in \mathbb{R}^{|\mathcal{S}|^2}$. Plugging this equality in the Bellman equation above we obtain

$$
\left\langle d_h^{\pi,\bar{\pi}}, r_h \right\rangle + \left\langle d_{h+1}^{\pi,\bar{\pi}}, E V_{h+1}^{\pi',\bar{\pi}} \right\rangle = \left\langle d_h^{\pi,\bar{\pi}}, Q_h^{\pi',\bar{\pi}} \right\rangle.
$$

Now, subtracting on both sides $\left\langle d_h^{\pi,\bar{\pi}}, EV_h^{\pi',\bar{\pi}} \right\rangle$ and rearranging, it holds that

$$\left\langle d_h^{\pi,\bar{\pi}}, r_h \right\rangle + \left\langle d_{h+1}^{\pi,\bar{\pi}}, EV_{h+1}^{\pi',\bar{\pi}} \right\rangle - \left\langle d_h^{\pi,\bar{\pi}}, EV_h^{\pi',\bar{\pi}} \right\rangle = \left\langle d_h^{\pi,\bar{\pi}}, Q_h^{\pi',\bar{\pi}} - EV_h^{\pi',\bar{\pi}} \right\rangle.$$

After this, taking sum from $h = 1$ to $H$ and recognizing that for all policy pairs $(\pi, \pi')$ it holds that $V_{H+1}^{\pi,\pi'} = 0$, it holds that

$$\sum_{h=1}^{H} \left\langle d_h^{\pi,\bar{\pi}}, r_h \right\rangle - \left\langle d_1^{\pi,\bar{\pi}}, EV_1^{\pi',\bar{\pi}} \right\rangle = \sum_{h=1}^{H} \left\langle d_h^{\pi,\bar{\pi}}, Q_h^{\pi',\bar{\pi}} - EV_h^{\pi',\bar{\pi}} \right\rangle.$$

Then, notice that for all policies $\pi, \bar{\pi}$ it holds that $\sum_{h=1}^{H} \left\langle d_h^{\pi,\bar{\pi}}, r_h \right\rangle = \langle \nu_1, V^{\pi,\bar{\pi}} \rangle$. Plugging in these observations, we get

$$\left\langle \nu_1, V^{\pi,\bar{\pi}} - V^{\pi',\bar{\pi}} \right\rangle = \sum_{h=1}^{H} \left\langle d_h^{\pi,\bar{\pi}}, Q_h^{\pi',\bar{\pi}} - EV_h^{\pi',\bar{\pi}} \right\rangle.$$

Therefore, expanding the expectation, and noticing that $d_h^{\pi,\bar{\pi}}(s, a, s', a' | s_1) = d_h^{\pi}(s, a | s_1) d_h^{\bar{\pi}}(s', a' | s_1)$ for all $h, s, a, s', a'$ and conditioning $s_1$, we get that

$$\left\langle \nu_1, V^{\pi,\bar{\pi}} - V^{\pi',\bar{\pi}} \right\rangle$$

$$= \mathbb{E}_{S_1 \sim \nu_1} \sum_{h=1}^{H} \mathbb{E}_{S \sim d_h^{\pi}|S_1} \left[ \left\langle \mathbb{E}_{s', A' \sim d_h^{\bar{\pi}}|S_1} Q_h^{\pi',\bar{\pi}}(S, \cdot, S', A'), \pi_h(\cdot|S, S_1) - \pi_h'(\cdot|S, S_1) \right\rangle \right].$$

$\square$

## D.2 Proof of Thm. 8

*Proof.* We set $\bar{\pi}_h^T(a_h|s_h) = \frac{\sum_{t=1}^{T} d_h^{\pi^t}(s_h, a_h)}{\sum_{t=1}^{T} d_h^{\pi^t}(s_h)}$, where $d(s)$ is the marginal distribution of $d(s, a)$ on state $s$, and $\bar{\pi}^T = (\bar{\pi}_h^T)_{h=1}^{H}$. We shows that $d_h^{\bar{\pi}^T} = \frac{1}{T} \sum_{t=1}^{T} d_h^{\pi^t}$ by induction. $h = 1$ holds by definition. Assuming on step $h$, the equation holds, we have

$$\begin{aligned}
d_{h+1}^{\bar{\pi}^T}(s_{h+1}, a_{h+1}) &= d_{h+1}^{\bar{\pi}^T}(s_{h+1}) \bar{\pi}_{h+1}^T(a_{h+1}|s_{h+1}) \\
&= \sum_{s_h, a_h \sim \bar{\pi}_h^T(\cdot|s_h)} d_h^{\bar{\pi}^T}(s_h, a_h) f(s_{h+1}|s_h, a_h) \bar{\pi}_{h+1}^T(a_{h+1}|s_{h+1}) \\
&= \sum_{s_h, a_h \sim \bar{\pi}_h^T(\cdot|s_h)} \frac{1}{T} \sum_{t=1}^{T} d_h^{\pi^t}(s_h, a_h) f(s_{h+1}|s_h, a_h) \bar{\pi}_{h+1}^T(a_{h+1}|s_{h+1}) \\
&= \frac{1}{T} \sum_{t=1}^{T} d_{h+1}^{\pi^t}(s_{h+1}) \bar{\pi}_{h+1}^T(a_{h+1}|s_{h+1}) \\
&= \frac{1}{T} \sum_{t=1}^{T} d_{h+1}^{\pi^t}(s_{h+1}, a_{h+1}),
\end{aligned}$$

where the last equation holds by definition of $\bar{\pi}_{h+1}^T$. Therefore, $h+1$ holds, and the $\bar{\pi}^T$ satisfy all equations for $h \in [H]$.

Using the value difference Lemma 1 we have that for any $\pi^\star \in \Pi$

$$\left\langle \nu_1, V^{\pi^\star,\pi^t} - V^{\pi^t,\pi^t} \right\rangle$$

$$= \mathbb{E}_{S_1 \sim \nu_1} \sum_{h=1}^{H} \mathbb{E}_{S \sim d_h^{\pi^\star}|S_1} \left[ \left\langle \mathbb{E}_{S', A' \sim d_h^{\pi^t}|S_1} Q_h^{\pi^t,\pi^t}(S, \cdot, S', A'), \pi_h^\star(\cdot|S) - \pi_h^t(\cdot|S) \right\rangle \right].$$

Therefore, summing over $t$ from $t = 1$ to $T$ we obtain

$$\sum_{t=1}^{T} \left\langle \nu_1, V^{\pi^\star, \pi^t} - V^{\pi^t, \pi^t} \right\rangle$$

$$= \mathbb{E}_{S_1 \sim \nu_1} \sum_{h=1}^{H} \mathbb{E}_{S \sim d_h^{\pi^\star} | S_1} \left[ \sum_{t=1}^{T} \left\langle \mathbb{E}_{S', A' \sim d_h^{\pi^t} | S_1} Q_h^{\pi^t, \pi^t}(S, \cdot, S', A'), \pi_h^\star(\cdot|S) - \pi_h^t(\cdot|S) \right\rangle \right].$$

Therefore, we need to control the local regrets at each state $s$ with loss $\ell_h^t(s, s_1) := -\mathbb{E}_{S', A' \sim d_h^{\pi^t} | s_1} Q_h^{\pi^t, \pi^t}(s, \cdot, S', A')$. To this end, we can invoke a standard convergence result for online mirror descent (Theorem 6.10 of Orabona (2023)) we obtain that at each state we have

$$\sum_{t=1}^{T} \left\langle \ell_h^t(s, s_1), \pi^\star(\cdot|s) - \pi^t(\cdot|s) \right\rangle \leq \frac{D(\pi^\star(\cdot|s), \pi^1(\cdot|s))}{\beta} + \beta \sum_{t=1}^{T} \|\ell_h^t(s, s_1)\|_\infty^2.$$

Now, noticing that we have $\|\ell_h^t(s, s_1)\|_\infty \leq H$ it holds that

$$\sum_{t=1}^{T} \left\langle \ell_h^t(s), \pi_h^\star(\cdot|s) - \pi_h^t(\cdot|s) \right\rangle \leq \frac{D(\pi_h^\star(\cdot|s), \pi_h^1(\cdot|s))}{\beta} + \beta T H^2.$$

Finally, using the assumption that $\pi^1(a|s) \geq \underline{\pi}$ for all $s, a \in \mathcal{S} \times \mathcal{A}$ it holds that $D(\pi^\star(\cdot|s), \pi^1(\cdot|s)) \leq \log \underline{\pi}^{-1}$. Therefore, choosing $\beta = \sqrt{\frac{\log \underline{\pi}^{-1}}{T H^2}}$ it holds that

$$\sum_{t=1}^{T} \left\langle \ell_h^t(s, s_1), \pi^\star(\cdot|s) - \pi^t(\cdot|s) \right\rangle \leq 2H \sqrt{T \log \underline{\pi}^{-1}}.$$

Thus, we conclude that

$$\sum_{t=1}^{T} \left\langle \nu_1, V^{\pi^\star, \pi^t} - V^{\pi^t, \pi^t} \right\rangle \leq 2H^2 \sqrt{T \log \underline{\pi}^{-1}}.$$

By the antisimmetry of the game, the same proof steps

$$\sum_{t=1}^{T} \left\langle \nu_1, V^{\pi^t, \pi^t} - V^{\pi^t, \bar{\pi}^\star} \right\rangle \leq 2H^2 \sqrt{T \log \underline{\pi}^{-1}}.$$

Therefore, it holds that for all $\pi^\star, \bar{\pi}^\star \in \Pi$

$$\sum_{t=1}^{T} \left\langle \nu_1, V^{\pi^\star, \pi^t} - V^{\pi^t, \pi^\star} \right\rangle \leq 4H^2 \sqrt{T \log \underline{\pi}^{-1}}.$$

Then, define $\bar{\pi}^T$ the trajectory level mixture policy as in Swamy et al. (2024), i.e. such that $d_h^{\bar{\pi}^T} = \frac{1}{T} \sum_{t=1}^{T} d_h^{\pi^t}$ for all stages $h \in [H]$. This implies that $V^{\bar{\pi}^T, \pi^\star} = \frac{1}{T} \sum_{t=1}^{T} V^{\pi^t, \pi^\star}$, and $V^{\pi^\star, \bar{\pi}^T} = \frac{1}{T} \sum_{t=1}^{T} V^{\pi^\star, \pi_t}$.

Therefore, we have that

$$\left\langle \nu_1, V^{\pi^\star, \bar{\pi}^T} - V^{\bar{\pi}^T, \bar{\pi}^\star} \right\rangle \leq 4H^2 \sqrt{\frac{\log \underline{\pi}^{-1}}{T}}.$$

Finally, selecting $\pi^\star = \left\langle \nu_1, \arg\max_{\pi \in \Pi} V^{\pi, \bar{\pi}^T} \right\rangle$ and $\bar{\pi}^\star = \left\langle \nu_1, \arg\min_{\pi \in \Pi} V^{\bar{\pi}^T, \pi} \right\rangle$, we obtain that

$$\max_{\pi \in \Pi} \left\langle \nu_1, V^{\pi, \bar{\pi}^T} \right\rangle - \min_{\pi \in \Pi} \left\langle \nu_1, V^{\bar{\pi}^T, \pi} \right\rangle \leq 4H^2 \sqrt{\frac{\log \underline{\pi}^{-1}}{T}}.$$

This implies that

$$\left\langle \nu_1, V^{\bar{\pi}^T, \bar{\pi}^T} \right\rangle - \min_{\pi \in \Pi} \left\langle \nu_1, V^{\bar{\pi}^T, \pi} \right\rangle \le 4H^2 \sqrt{\frac{\log \pi^{-1}}{T}},$$

and

$$\max_{\pi \in \Pi} \left\langle \nu_1, V^{\pi, \bar{\pi}^T} \right\rangle - \left\langle \nu_1, V^{\bar{\pi}^T, \bar{\pi}^T} \right\rangle \le 4H^2 \sqrt{\frac{\log \pi^{-1}}{T}},$$

Therefore, setting $T = \frac{16H^4 \log \pi^{-1}}{\epsilon^2}$ we obtain an $\epsilon$-approximate Nash equilibrium. $\qquad\square$

### D.3 Proof of Theorem 4

*Proof.* The optimization problem

$$\underset{d \in \tilde{\mathcal{F}}}{\arg\max} \min_{d' \in \tilde{\mathcal{F}}} \mathbb{E}_{s_1 \sim \nu_1} \sum_{h=1}^{H} \sum_{s,a,s',a'} d_h(s,a|s_1) r(s,a,s',a') d'_h(s',a'|s_1)$$

can be carried out individually over possible initial states. That is for each $s_1 \in \mathrm{supp}(\nu_1)$ we aim at solving

$$\underset{d \in \mathcal{F}_{s_1}}{\arg\max} \min_{d' \in \mathcal{F}_{s_1}} \sum_{h=1}^{H} \sum_{s,a,s',a'} d_h(s,a|s_1) r(s,a,s',a') d'_h(s',a'|s_1)$$

To this end for any $s_1$, we consider $\phi_h^t \in \mathcal{F}$ and $\psi_h^t \in \mathcal{F}$ which are generated by the following updates

$$\phi_h^{t+1} = \underset{\phi \in \mathcal{F}_{s_1}}{\arg\max} \beta \left\langle \phi, 2\mathbb{E}_{s',a' \sim \psi^t} r_h(\cdot, \cdot, s', a') - \mathbb{E}_{s',a' \sim \psi^{t-1}} r_h(\cdot, \cdot, s', a') \right\rangle - \mathbb{D}(\phi, \phi_h^t),$$

and

$$\psi_h^{t+1} = \underset{\psi \in \mathcal{F}_{s_1}}{\arg\min} \beta \left\langle \psi, 2\mathbb{E}_{s',a' \sim \phi^t} r_h(s', a', \cdot, \cdot) - \mathbb{E}_{s',a' \sim \phi^{t-1}} r_h(s', a', \cdot, \cdot) \right\rangle + \mathbb{D}(\psi, \psi_h^t),$$

In order to prove convergence to an $\epsilon$-approximate Nash equilibrium, we need to control the quantity

$$\mathrm{Gap}_{s_1} = \frac{1}{T} \sum_{h=1}^{H} \sum_{t=1}^{T} \left\langle \theta_h^t, \phi_h^\star - \phi_h^t \right\rangle + \frac{1}{T} \sum_{h=1}^{H} \sum_{t=1}^{T} \left\langle \zeta_h^t, \psi_h^\star - \psi_h^t \right\rangle,$$

for $\theta_h^t(s,a) = \sum_{s',a'} \psi_h^t(s',a') r_h(s,a,s',a')$ and $\zeta_h^t(s',a') = -\sum_{s,a} \phi_h^t(s,a) r_h(s,a,s',a')$. At this point, we bound the local regret term with the OMPO update. We have that for any $\phi_h \in \mathcal{F}$

$$\begin{aligned}
\beta \left\langle 2\theta_h^t - \theta_h^{t-1}, \phi_h - \phi_h^{t+1} \right\rangle &= \beta \left\langle \theta_h^t - \theta_h^{t+1}, \phi_h - \phi_h^{t+1} \right\rangle \\
&\quad + \beta \left\langle \theta_h^t + \theta_h^{t+1} - \theta_h^{t-1}, \phi_h - \phi_h^{t+1} \right\rangle \\
&= \beta \left\langle \theta_h^t - \theta_h^{t+1}, \phi_h - \phi_h^{t+1} \right\rangle \\
&\quad + \beta \left\langle \theta_h^t - \theta_h^{t-1}, \phi_h - \phi_h^{t} \right\rangle \\
&\quad + \beta \left\langle \theta_h^t - \theta_h^{t-1}, \phi_h^t - \phi_h^{t+1} \right\rangle \\
&\quad + \beta \left\langle \theta_h^{t+1}, \phi_h - \phi_h^{t+1} \right\rangle.
\end{aligned}$$

At this point, we work on the third summand above

$$-\beta \left\langle \theta_h^t - \theta_h^{t-1}, \phi_h^t - \phi_h^{t+1} \right\rangle \le \beta^2 \lambda \|\theta_h^t - \theta_h^{t-1}\|_\infty^2 + \frac{1}{4\lambda} \|\phi_h^t - \phi_h^{t+1}\|_1^2.$$

In addition, we have that $\|\theta_h^t - \theta_h^{t-1}\|_\infty \le \|\psi_h^t - \psi_h^{t-1}\|_1$ and we can apply the $1/\lambda$ strong convexity of $\mathbb{D}$, we obtain

$$\beta \left\langle \theta_h^t - \theta_h^{t-1}, \phi_h^t - \phi_h^{t+1} \right\rangle \le \lambda \beta^2 \|\psi_h^t - \psi_h^{t-1}\|_1^2 + \frac{1}{2}\mathbb{D}(\phi_h^{t+1}, \phi_h^t).$$

On the other hand, by the three point identity we have that for all $\phi \in \mathcal{F}$

$$\mathbb{D}(\phi_h, \phi_h^{t+1}) = \mathbb{D}(\phi_h, \phi_h^t) - \mathbb{D}(\phi_h^{t+1}, \phi_h^t) + \left\langle \nabla \mathbb{D}(\phi_h^{t+1}, \phi_h^t), \phi_h^{t+1} - \phi_h \right\rangle$$

Then, using the property of the update rule, we obtain that

$$\left\langle \nabla \mathbb{D}(\phi_h^{t+1}, \phi_h^t), \phi_h^{t+1} - \phi_h \right\rangle \leq \beta \left\langle 2\theta_h^t - \theta_h^{t-1}, \phi_h^{t+1} - \phi_h \right\rangle.$$

Putting all the pieces together we have that

$$\begin{aligned}
\mathbb{D}(\phi_h, \phi_h^{t+1}) \leq{}& \mathbb{D}(\phi_h, \phi_h^t) - \mathbb{D}(\phi_h^{t+1}, \phi_h^t) + \beta \left\langle 2\theta_h^t - \theta_h^{t-1}, \phi_h^{t+1} - \phi_h \right\rangle \\
\leq{}& \mathbb{D}(\phi_h, \phi_h^t) - \mathbb{D}(\phi_h^{t+1}, \phi_h^t) \\
& - \beta \left\langle \theta_h^t - \theta_h^{t+1}, \phi_h - \phi_h^{t+1} \right\rangle \\
& - \beta \left\langle \theta_h^t - \theta_h^{t-1}, \phi_h - \phi_h^t \right\rangle \\
& + \beta^2 \lambda \|\psi_h^t - \psi_h^{t-1}\|_1^2 + \frac{1}{2} \mathbb{D}(\phi_h^{t+1}, \phi_h^t) \\
& - \beta \left\langle \theta_h^{t+1}, \phi_h - \phi_h^{t+1} \right\rangle.
\end{aligned}$$

Now, rearranging the terms we get

$$\begin{aligned}
\beta \left\langle \theta_h^{t+1}, \phi_h - \phi_h^{t+1} \right\rangle \leq{}& \mathbb{D}(\phi_h, \phi_h^t) - \mathbb{D}(\phi_h, \phi_h^{t+1}) - \frac{1}{2} \mathbb{D}(\phi_h^{t+1}, \phi_h^t) \\
& - \beta \left\langle \theta_h^t - \theta_h^{t+1}, \phi_h - \phi_h^{t+1} \right\rangle \\
& - \beta \left\langle \theta_h^t - \theta_h^{t-1}, \phi_h - \phi_h^t \right\rangle \\
& + \beta^2 \lambda \|\psi_h^t - \psi_h^{t-1}\|_1^2.
\end{aligned}$$

Now, denoting $\Phi_\phi^t := \mathbb{D}(\phi_h, \phi_h^t) - \beta \left\langle \theta_h^t - \theta_h^{t-1}, \phi_h - \phi_h^t \right\rangle$ and summing over $t$ we obtain

$$\beta \sum_{t=1}^T \left\langle \theta_h^t, \phi_h - \phi_h^t \right\rangle \leq \sum_{t=1}^T \Phi_\phi^{t-1} - \Phi_\phi^t - \frac{1}{2} \sum_{t=1}^T \mathbb{D}(\phi_h^t, \phi_h^{t-1}) + \beta^2 \lambda \sum_{t=1}^T \|\psi_h^{t-1} - \psi_h^{t-2}\|_1^2.$$

Similarly we get

$$\beta \sum_{t=1}^T \left\langle \zeta^t, \psi_h - \psi_h^t \right\rangle \leq \sum_{t=1}^T \Phi_\psi^{t-1} - \Phi_\psi^t - \frac{1}{2} \sum_{t=1}^T \mathbb{D}(\psi_h^t, \psi_h^{t-1}) + \beta^2 \lambda \sum_{t=1}^T \|\phi_h^{t-1} - \phi_h^{t-2}\|_1^2.$$

Now, using $1/\lambda$ strong convexity of $\mathbb{D}$ and summing the two terms we have that

$$\begin{aligned}
\beta T \mathrm{Gap}_{s_1, h} \leq{}& \Phi^0 - \Phi^{T-1} - \frac{1}{2} \sum_{t=1}^T (\mathbb{D}(\psi_h^t, \psi_h^{t-1}) + \mathbb{D}(\phi_h^t, \phi_h^{t-1})) \\
& + 2\beta^2 \lambda \sum_{t=1}^T (\mathbb{D}(\psi_h^{t-1}, \psi_h^{t-2}) + \mathbb{D}(\phi_h^{t-1}, \phi_h^{t-2})),
\end{aligned}$$

with $\Phi^t = \Phi_\phi^t + \Phi_\psi^t$. At this point, setting $\beta \leq \frac{1}{\sqrt{2\lambda}}$, we obtain a telescopic sum

$$\begin{aligned}
& \beta T \mathrm{Gap}_{s_1, h} \\
\leq{}& \Phi^0 - \Phi^{T-1} - \frac{1}{2} \sum_{t=1}^T (\mathbb{D}(\psi_h^t, \psi_h^{t-1}) + \mathbb{D}(\phi_h^t, \phi_h^{t-1}) - \mathbb{D}(\psi_h^{t-1}, \psi_h^{t-2}) - \mathbb{D}(\phi_h^{t-1}, \phi_h^{t-2})) \\
\leq{}& \Phi^0 - \Phi^{T-1} + \frac{1}{2} \left( \mathbb{D}(\psi_h^1, \psi_h^0) + \mathbb{D}(\phi_h^1, \phi_h^0) \right).
\end{aligned}$$

Now recalling that by assumption the occupancy measure of the reference policy is lower bounded, i.e. $d^{\pi^1} \geq \underline{d}$, we can upper bound $\Phi^0 - \Phi^T \leq 2\log\underline{d}^{-1} + 8\beta$ that allows to conclude that for all $n \in [N]$ and setting $\psi_h^0 = \psi_h^1$ and $\phi_h^1 = \phi_h^0$,

$$\text{Gap}_{s_1,h} \leq \frac{2\log\underline{d}^{-1} + 8\beta}{\beta T} \leq \frac{10\log\underline{d}^{-1}}{\beta T}.$$

Now, notice that Gap can be rewritten as

$$
\begin{aligned}
\text{Gap}_{s_1} &= \sum_{h=1}^{H} \text{Gap}_{s_1,h} \\
&= \frac{1}{T} \sum_{t=1}^{T} \sum_{h=1}^{H} \sum_{s,a,s',a'} \psi_h^t(s',a')r_h(s,a,s',a')\phi_h^\star(s,a) \\
&\qquad - \frac{1}{T} \sum_{t=1}^{T} \sum_{h=1}^{H} \sum_{s,a,s',a'} \psi_h^\star(s',a')r_h(s,a,s',a')\phi_h^t(s,a) \\
&= \sum_{h=1}^{H} \sum_{s,a,s',a'} \frac{1}{T} \sum_{t=1}^{T} \psi_h^t(s',a')r_h(s,a,s',a')\phi_h^\star(s,a) \\
&\qquad - \sum_{h=1}^{H} \sum_{s,a,s',a'} \psi_h^\star(s',a')r_h(s,a,s',a')\frac{1}{T} \sum_{t=1}^{T} \phi_h^t(s,a) \\
&= \sum_{h=1}^{H} \sum_{s,a,s',a'} \bar{\psi}_h(s',a')r_h(s,a,s',a')\phi_h^\star(s,a) - \sum_{h=1}^{H} \sum_{s,a,s',a'} \psi_h^\star(s',a')r_h(s,a,s',a')\bar{\phi}_h(s,a).
\end{aligned}
$$

At this point, let us define $\pi_\phi^{\text{out}}(a|s) = \frac{\bar{\phi}(s,a)}{\sum_a \bar{\phi}(s,a)}$ and $\pi_\psi^{\text{out}}(a|s) = \frac{\bar{\psi}(s,a)}{\sum_a \bar{\psi}(s,a)}$. For such policies and by appropriate choice for $\psi^\star$ and $\phi^\star$ it follows that

$$\text{Gap}_{s_1} = \max_\phi V^{\phi,\pi_\psi^{\text{out}}}(s_1) - \min_\psi V^{\pi_\phi^{\text{out}},\psi}(s_1).$$

By the bound on $\text{Gap}_{s_1}$ for each $s_1 \in \text{supp}(\nu_1)$, it follows that

$$\left\langle \nu_1, \max_\phi V^{\phi,\pi_\psi^{\text{out}}} - \min_\psi V^{\pi_\phi^{\text{out}},\psi} \right\rangle = \mathbb{E}_{s_1 \sim \nu_1} \text{Gap}_{s_1} \leq \frac{10H\log\underline{d}^{-1}}{\beta T},$$

therefore $T \geq \frac{10H\log\underline{d}^{-1}}{\beta\epsilon}$. The proof is concluded invoking Thm. 5 that ensures that the policies $\pi_\psi^{\text{out}}$ and $\pi_\phi^{\text{out}}$ coincide. $\qquad\square$

### D.4 Proof of Theorem 5

*Proof.* Let us consider two players performing the following updates

$$\phi_h^{t+1} = \arg\max_{\phi \in \mathcal{F}_{s_1}} \beta \left\langle \phi, 2\mathbb{E}_{s',a'\sim\psi^t} r_h(\cdot,\cdot,s',a') - \mathbb{E}_{s',a'\sim\psi^{t-1}} r_h(\cdot,\cdot,s',a') \right\rangle - \mathbb{D}(\phi,\phi_h^t),$$

and

$$\psi_h^{t+1} = \arg\min_{\psi \in \mathcal{F}_{s_1}} \beta \left\langle \psi, 2\mathbb{E}_{s',a'\sim\phi^t} r_h(s',a',\cdot,\cdot) - \mathbb{E}_{s',a'\sim\phi^{t-1}} r_h(s',a',\cdot,\cdot) \right\rangle + \mathbb{D}(\psi,\psi_h^t).$$

The goal is to proof that the iterates generated by the two updates are identical. We will prove this fact by induction. The base case holds by initialization which gives $\phi_h^0 = \psi_h^0$ for all $h \in [H]$. Then, let us assume by

the induction step that $\psi_h^t = \phi_h^t$ for all $h \in [H]$, then

$$\phi_h^{t+1}$$
$$= \underset{\phi \in \mathcal{F}_{s_1}}{\arg\max} \; \beta \left\langle \phi, 2\mathbb{E}_{s',a' \sim \psi^t} r_h(\cdot, \cdot, s', a') - \mathbb{E}_{s',a' \sim \psi^{t-1}} r_h(\cdot, \cdot, s', a') \right\rangle - \mathbb{D}(\phi, \phi_h^t)$$
$$= \underset{\phi \in \mathcal{F}_{s_1}}{\arg\max} \; \beta \left\langle \phi, -2\mathbb{E}_{s',a' \sim \psi^t} r_h(s', a', \cdot, \cdot) + \mathbb{E}_{s',a' \sim \psi^{t-1}} r_h(s', a', \cdot, \cdot) \right\rangle - \mathbb{D}(\phi, \phi_h^t) + \beta \left\langle \phi, \mathbf{1} \right\rangle$$

(Antisymmetric Reward)
$$= \underset{\phi \in \mathcal{F}_{s_1}}{\arg\max} \; \beta \left\langle \phi, -2\mathbb{E}_{s',a' \sim \psi^t} r_h(s', a', \cdot, \cdot) + \mathbb{E}_{s',a' \sim \psi^{t-1}} r_h(s', a', \cdot, \cdot) \right\rangle - \mathbb{D}(\phi, \phi_h^t) + \beta$$

(Normalization of $\phi$)
$$= \underset{\phi \in \mathcal{F}_{s_1}}{\arg\max} \; \beta \left\langle \phi, -2\mathbb{E}_{s',a' \sim \psi^t} r_h(s', a', \cdot, \cdot) + \mathbb{E}_{s',a' \sim \psi^{t-1}} r_h(s', a', \cdot, \cdot) \right\rangle - \mathbb{D}(\phi, \phi_h^t)$$

($\beta$ does not depend on $\phi$)
$$= \underset{\phi \in \mathcal{F}_{s_1}}{\arg\max} \; \beta \left\langle \phi, -2\mathbb{E}_{s',a' \sim \phi^t} r_h(s', a', \cdot, \cdot) + \mathbb{E}_{s',a' \sim \phi^{t-1}} r_h(s', a', \cdot, \cdot) \right\rangle - \mathbb{D}(\phi, \psi_h^t)$$

(Inductive Hypothesis)
$$= \underset{\psi \in \mathcal{F}_{s_1}}{\arg\min} \; \beta \left\langle \psi, 2\mathbb{E}_{s',a' \sim \phi^t} r_h(s', a', \cdot, \cdot) - \mathbb{E}_{s',a' \sim \phi^{t-1}} r_h(s', a', \cdot, \cdot) \right\rangle + \mathbb{D}(\psi, \psi_h^t)$$

(Renaming the optimization variable and $\underset{x}{\arg\max} \, f(x) = \underset{x}{\arg\min} -f(x)$)
$$= \psi_h^{t+1}.$$

$\square$

## D.5 Proof of Thm. 6

*Proof.* As we assumed that $d^\star \geq d_{\min} > 0$, let us modify the updates projecting onto $\mathcal{F} \cap \{d \in \mathcal{F} : d \geq d_{\min}\}$. This makes the negative entropy differentiable over the whole domain. The first step is to establish summability of the iterates difference in the squared norm. To this end, let us recall that we proved along the proof of Thm. 4 that

$$\beta \sum_{t=1}^{T} \left\langle \theta_h^t, \phi_h - \phi_h^t \right\rangle \leq \sum_{t=1}^{T} \Phi_\phi^{t-1} - \Phi_\phi^t - \frac{1}{2} \sum_{t=1}^{T} \mathbb{D}(\phi_h^t, \phi_h^{t-1}) + \beta^2 \lambda \sum_{t=1}^{T} \|\psi_h^{t-1} - \psi_h^{t-2}\|_1^2.$$

and

$$\beta \sum_{t=1}^{T} \left\langle \zeta^t, \psi_h - \psi_h^t \right\rangle \leq \sum_{t=1}^{T} \Phi_\psi^{t-1} - \Phi_\psi^t - \frac{1}{2} \sum_{t=1}^{T} \mathbb{D}(\psi_h^t, \psi_h^{t-1}) + \beta^2 \lambda \sum_{t=1}^{T} \|\phi_h^{t-1} - \phi_h^{t-2}\|_1^2.$$

where $\Phi_\phi^t := \mathbb{D}(\phi_h, \phi_h^t) - \beta \left\langle \theta_h^t - \theta_h^{t-1}, \phi_h - \phi_h^t \right\rangle$ and $\Phi_\psi^t := \mathbb{D}(\psi_h, \psi_h^t) - \beta \left\langle \zeta_h^t - \zeta_h^{t-1}, \psi_h - \psi_h^t \right\rangle$ and $\Phi^t = \Phi_\phi^t + \Phi_\psi^t$. Summing the two above inequalities and the $1/\lambda$ strong convexity of the Bregman divergence we obtain[6]

$$\beta \sum_{t=1}^{T} \left\langle \theta_h^t, \phi_h - \phi_h^t \right\rangle + \beta \sum_{t=1}^{T} \left\langle \zeta^t, \psi_h - \psi_h^t \right\rangle \leq \Phi^1 - \Phi^T \tag{5}$$

$$- \left( \frac{1}{4\lambda} - \beta^2 \lambda \right) \sum_{t=1}^{T} \left( \|\phi_h^{t-1} - \phi_h^{t-2}\|_1^2 + \|\psi_h^{t-1} - \psi_h^{t-2}\|_1^2 \right). \tag{6}$$

---

[6]We also used that $\mathbb{D}(\phi^T, \phi^{T-1}) + \mathbb{D}(\psi^T, \psi^{T-1}) \geq 0$ and that $\phi_h^0 = \phi_0^{-1}$ by initialization.

As in the proof of Thm. 4, we can set $\phi_h = \phi_h^\star$ and $\psi_h = \psi_h^\star$ to ensure that the LHS is positive and we upper bound $\Phi^0 - \Phi^T \le 2 \log \underline{d}^{-1} + 8\beta$. We obtain

$$\left(\frac{1}{4\lambda} - \beta^2 \lambda\right) \sum_{t=1}^T \left(\|\phi_h^{t-1} - \phi_h^{t-2}\|_1^2 + \|\psi_h^{t-1} - \psi_h^{t-2}\|_1^2\right) \le 2 \log \underline{d}^{-1} + 8\beta$$

Therefore for $\beta \le 1/\sqrt{8\lambda^2}$, we have that

$$\sum_{t=1}^T \left(\|\phi_h^{t-1} - \phi_h^{t-2}\|_1^2 + \|\psi_h^{t-1} - \psi_h^{t-2}\|_1^2\right) \le 16\lambda \log \underline{d}^{-1} + 64\lambda\beta$$

Therefore the sequence of the iterates difference squared is summable. Moreover, since the iterates belongs to a closed compact set there exists a subsequence $\left\{\phi_h^{t_n}, \psi_h^{t_n}\right\}_{n=1}^\infty$ which converges to $\{\phi_h^\infty, \psi_h^\infty\}$ for all $h \in [H]$. Moreover the fact that the iterates difference squared is summable implies that

$$\lim_{t\to\infty} \|\phi_h^{t-1} - \phi_h^{t-2}\|_1^2 + \|\psi_h^{t-1} - \psi_h^{t-2}\|_1^2 = 0$$

Therefore, for the convergent subsequence it holds that

$$\lim_{t\to\infty} \|\phi_h^{t_n-1} - \phi_h^{t_n}\|_1^2 + \|\psi_h^{t_n} - \psi_h^{t_n-1}\|_1^2 = 0$$

Therefore the subsequences $\left\{\phi_h^{t_n}, \psi_h^{t_n}\right\}_{n=1}^\infty$ and $\left\{\phi_h^{t_n-1}, \psi_h^{t_n-1}\right\}_{n=1}^\infty$ both converge to $\{\phi_h^\infty, \psi_h^\infty\}$. At this point, notice that our update rule implies that

$$\left\langle 2\theta_h^{t_n} - \theta_h^{t_n-1} + \nabla\omega(\phi_h^{t_n+1}) - \nabla\omega(\phi_h^{t_n}), \phi_h - \phi_h^{t_n+1}\right\rangle \le 0 \quad \forall h \in [H], \forall \phi_h$$

and

$$\left\langle 2\zeta_h^{t_n} - \zeta_h^{t_n-1} + \nabla\omega(\psi_h^{t_n+1}) - \nabla\omega(\psi_h^{t_n}), \psi_h - \psi_h^{t_n+1}\right\rangle \le 0 \quad \forall h \in [H], \forall \psi_h$$

where $\omega$ denotes the potential function inducing the Bregman divergence $\mathbb{D}$. That is, $\mathbb{D}(x, y) = \omega(x) - \omega(y) - \langle\nabla\omega(y), x - y\rangle$. At this point, the fact that $\omega$ is continuous differentiable over the whole domain $\mathcal{F} \cap \{d \in \mathcal{F} : d \ge d_{\min}\}$ it holds that

$$\langle\theta_h^\infty, \phi_h - \phi_h^\infty\rangle \le 0 \quad \forall h \in [H], \forall\phi_h$$

and

$$\langle\zeta_h^\infty, \psi_h - \psi_h^\infty\rangle \le 0 \quad \forall h \in [H], \forall\psi_h$$

Therefore, $\phi^\infty, \psi^\infty$ ( the limit of the subsequence ) is a Nash equilibrium point.

At this point, to establish convergence of the sequence let us notice that rearranging Equation 6 ( and not considering the sum over $t$), it holds that

$$\beta\left\langle\theta_h^t, \phi_h - \phi_h^t\right\rangle + \left\langle\zeta^t, \psi_h - \psi_h^t\right\rangle \le \Phi^{t-1} - \Phi^t$$

$$- \left(\frac{1}{4\lambda} - \beta^2\lambda\right)\left(\|\phi_h^{t-1} - \phi_h^{t-2}\|_1^2 + \|\psi_h^{t-1} - \psi_h^{t-2}\|_1^2\right).$$

$$= E^{t-1} - E^t + \beta L^{t-1} - \beta L^t - \left(\frac{1}{4\lambda} - \beta^2\lambda\right)\left(\|\phi_h^{t-1} - \phi_h^{t-2}\|_1^2 + \|\psi_h^{t-1} - \psi_h^{t-2}\|_1^2\right).$$

where the last line introduced the notation $E^t = \mathbb{D}(\phi_h, \phi_h^t) + \mathbb{D}(\psi_h, \psi_h^t)$ and $L^t = -\left\langle\theta_h^t - \theta_h^{t-1}, \phi_h - \phi_h^t\right\rangle - \left\langle\zeta_h^t - \zeta_h^{t-1}, \psi_h - \psi_h^t\right\rangle$ and we used the fact that $\Phi^t = E^t + L^t$. At this point, choosing $\phi_h = \phi_h^\star$ and $\psi_h = \psi_h^\star$ we have that the LHS is zero and $L^t$ is summable. Indeed,

$$\sum_{t=1}^T L^t = \sum_{t=1}^T \left(-\left\langle\theta_h^t - \theta_h^{t-1}, \phi_h - \phi_h^t\right\rangle - \left\langle\zeta_h^t - \zeta_h^{t-1}, \psi_h - \psi_h^t\right\rangle\right)$$

$$= \sum_{t=1}^T \left(-\left\langle\theta_h^t - \theta_h^{t-1}, \phi_h\right\rangle - \left\langle\zeta_h^t - \zeta_h^{t-1}, \psi_h\right\rangle\right)$$

$$= \left\langle\theta_h^0 - \theta_h^T, \phi_h\right\rangle + \left\langle\zeta_h^0 - \zeta_h^T, \psi_h\right\rangle$$

$$\le 2.$$

Therefore, we can rearrange and obtain

$$E^t \le E^{t-1} + \beta L^{t-1} - \beta L^t - \left( \frac{1}{4\lambda} - \beta^2 \lambda \right) \left( \|\phi_h^{t-1} - \phi_h^{t-2}\|_1^2 + \|\psi_h^{t-1} - \psi_h^{t-2}\|_1^2 \right)$$

Therefore, $E^t$ is a quasi-Féjer sequence and hence it has a limit $E^\infty$. At this point, we can notice that $0 = \lim_{n\to\infty} \|\phi_h^{t_n} - \phi_h^\star\| + \|\psi_h^{t_n} - \psi_h^\star\|$ by the convergence of the subsequence, implies that $\lim_{n\to\infty} \mathbb{D}(\phi_h^{t_n}, \phi_h^\star) + \mathbb{D}(\psi_h^{t_n}, \psi_h^\star) = 0$ by the reciprocity condition which holds since our constraints define a subset of the simplex. However, by convergence of the energy levels, $\lim_{t\to\infty} \mathbb{D}(\phi_h^t, \phi_h^\star) + \mathbb{D}(\psi_h^t, \psi_h^\star)$ exists and must be equal to the limit of the subsequence. Therefore, $\lim_{t\to\infty} \mathbb{D}(\phi_h^t, \phi_h^\star) + \mathbb{D}(\psi_h^t, \psi_h^\star) = 0$. Finally by strong convexity of the Bregman divergence we can conclude $\lim_{t\to\infty} \|\phi_h^t - \phi_h^\star\|_1^2 + \|\psi_h^t - \psi_h^\star\|_1^2 = 0$. □

## E   Implementation of Algorithm 1 with updates over policies

In this section, we explain how the update in Algorithm 1 for different choices of $\mathbb{D}$. In both cases, we will derive an update that can be summarized by following template. Let us define $r_h^t(s,a) = \mathbb{E}_{s',a'\sim d_h^t} r(s,a,s',a')$ and $r_h^{t-1}(s,a) = \mathbb{E}_{s',a'\sim d_h^{t-1}} r(s,a,s',a')$

- Compute the $Q_h^t$ function corresponding to the reward function $2r_h^t - r_h^{t-1}$ minimizing a loss function that depends on the choice of $\mathbb{D}$.

- Update the policy as

$$\pi_h^{t+1}(a|s) \propto \pi_h^t(a|s) \exp\left( \beta Q_h^t(s,a) \right).$$

Finally, in Appx. E.3 we show that for $\mathbb{D}$ being the conditional relative entropy and for $\beta$ small enough the value function $Q_h^t$ is well approximated by the standard Bellman equations.

**Remark 3.** *Both choices of the Bregman divergence are $1$ strongly convex so Thm. 4 applies with $\lambda = 1$.*

In the following we consider a generic reward function $\tilde{r}$. In our setting, we will apply the following results for $\tilde{r}_h^t = 2r_h^t - r_h^{t-1}$ in order to implement the updates of Alg. 1 for the different values of $h$ and $t$.

### E.1   $\mathbb{D}$ chosen as the sum of conditional and relative entropy

In this section, we explain how to implement the occupancy measure update in Algorithm 1 over policies. We use the machinery for single agent MDPs introduced in Bas-Serrano et al. (2021). In particular, we consider the Bregman divergence given by the sum of the relative entropy $D(d,d') = \sum_{s,a} d(s,a) \log\left( \frac{d(s,a)}{d'(s,a)} \right)$ and of the conditional relative entropy given, i.e. $H(d,d') = \sum_{s,a} d(s,a) \log\left( \frac{\pi_d(a|s)}{\pi_{d'}(a|s)} \right)$ with $\pi_d(a|s) = d(s,a)/\sum_a d(s,a)$. Under this choice for $\mathbb{D}$, the update of Algorithm 1 for particular values of $h, t, s_1$ corresponds to the solution of the following optimization program

$$d_h^{t+1} = \arg\max_{d\in\Delta^H} \sum_{h=1}^H \langle d_h, \tilde{r}_h^t \rangle - \frac{1}{\beta} D(d_h, d_h^t) - \frac{1}{\beta} H(d_h, d_h^t),$$
$$\text{s.t.} \quad E^T d_h = F^T d_{h-1} \quad \forall h \in [H]. \tag{Update I}$$

**Theorem 9.** *The policy $\pi_h^{t+1}$ with occupancy measure $d_h^{t+1}$ defined in Eq. (Update I) can be computed as follows*

$$\pi_h^{t+1}(a|s) \propto \pi_h^t(a|s) \exp\left( \beta Q_h^t(s,a) \right),$$

*where $Q_h^t$ is the minimizer of the following loss*

$$\frac{1}{\beta} \sum_{h=1}^H \log \sum_{s,a} \mu_h^t(s,a) \exp\left( \beta(2\tilde{r}_h^t + PV_{h+1} - Q_h)(s,a) \right) + \langle \nu_1, V_1 \rangle,$$

*while $V_{h+1}^t$ is given by the following closed form.*

$$V_{h+1}^t(s) = \frac{1}{\beta} \log \sum_a \pi_h^t(a|s) \exp(\beta Q_{h+1}^t(s,a)).$$

*Proof.* Let us introduce an auxiliary variable $\mu_h = d_h$ for all $h \in [H]$, then we can rewrite the optimization program as

$$\arg\max_{d \in \Delta^H} \max_{\mu \in \Delta^H} \sum_{h=1}^H \langle \mu_h, \tilde{r}_h \rangle - \frac{1}{\beta} D(\mu_h, \mu_h^t) - \frac{1}{\beta} H(d_h, d_h^t),$$

$$\text{s.t.} \quad E^T d_h = F^T \mu_{h-1} \quad \forall h \in [H],$$

$$\text{s.t.} \quad \mu_h = d_h \quad \forall h \in [H].$$

Then, by Lagrangian duality we have that

$$\max_{d \in \Delta^H} \max_{\mu \in \Delta^H} \min_{Q,V} \sum_{h=1}^H \langle \mu_h, \tilde{r} \rangle - \frac{1}{\beta} D(\mu_h, \mu_h^t) - \frac{1}{\beta} H(d_h, d_h^t)$$

$$+ \left\langle -E^T d_h + F^T \mu_{h-1}, V_h \right\rangle + \langle Q_h, d_h - \mu_h \rangle$$

$$= \max_{d \in \Delta^H} \max_{\mu \in \Delta^H} \min_{Q,V} \sum_{h=1}^H \langle \mu_h, \tilde{r} + FV_{h+1} - Q_h \rangle + \langle d_h, Q_h - EV_h \rangle$$

$$- \frac{1}{\beta} D(\mu_h, \mu_h^t) - \frac{1}{\beta} H(d_h, d_h^t)$$

$$+ \langle \nu_1, V_1 \rangle = \mathcal{L}^\star.$$

Then, by Lagrangian duality, we have that the objective is unchanged by swapping the min and max

$$\mathcal{L}^\star = \min_{Q,V} \max_{d \in \Delta^H} \max_{\mu \in \Delta^H} \sum_{h=1}^H \langle \mu_h, \tilde{r}_h + FV_{h+1} - Q_h \rangle + \langle d_h, Q_h - EV_h \rangle$$

$$- \frac{1}{\beta} D(\mu_h, \mu_h^t) - \frac{1}{\beta} H(d_h, d_h^t) + \langle \nu_1, V_1 \rangle .$$

The inner maximization is solved by the following values

$$\mu_h^+(Q,V) \propto \mu_h^t \odot \exp\left(\beta(\tilde{r}_h + FV_{h+1} - Q_h)\right),$$

$$\pi_h^+(Q,V;s) \propto \pi_h^t(\cdot|s) \odot \exp\left(\beta(Q_h(s,\cdot) - V_h(s))\right),$$

where $\odot$ denotes the elementwise product between vectors. Then, replacing these values in the Lagrandian and parameterizing the functions $V_h$ by the functions $Q_h$ to ensure normalization of the policy, i.e. $V_h(s) = \frac{1}{\beta} \log \sum_a \pi_h^t(a|s) \exp(\beta Q_h(s,a))$ we have that

$$\mathcal{L}^\star = \min_Q \frac{1}{\beta} \sum_{h=1}^H \log \sum_{s,a} \mu_h^t(s,a) \exp\left(\beta(\tilde{r}_h + FV_{h+1} - Q_h)(s,a)\right) + \langle \nu_1, V_1 \rangle .$$

Therefore, denoting

$$Q_h^t = \arg\min_Q \frac{1}{\beta} \sum_{h=1}^H \log \sum_{s,a} \mu_h^t(s,a) \exp\left(\beta(\tilde{r}_h + FV_{h+1} - Q_h)(s,a)\right) + \langle \nu_1, V_1 \rangle ,$$

and $V_h^t = \frac{1}{\beta} \log \sum_a \pi_h^t(a|s) \exp(\beta Q_h^t(s,a))$, we have that the policy $\pi_h^{t+1}(\cdot|s) = \pi_h^+(Q^t, V^t; s)$ has occupancy measure equal to $d_h^{t+1}$ for all $h \in [H]$. This is because by the constraints of the problem we have that $d_h^{t+1}$ satisfies the Bellman flow constraints and that the policy $\pi_h^{t+1}$ satisfies $\pi_h^{t+1}(a|s) = d_h^t(s,a)/\sum_a d_h^t(s,a)$. $\qquad\square$

### E.2 $\mathbb{D}$ chosen as conditional relative entropy Neu et al. (2017)

In this section, we study the update considering $\mathbb{D}$ chosen as sum of the conditional relative entropy over the stages $h'$ s.t. $1 \leq h' \leq h$, i.e. we study the following update.[7]

$$d^{t+1} = \underset{d \in \Delta^H}{\arg\max} \sum_{h=1}^{H} \left( \langle d_h, \tilde{r}_h \rangle - \frac{1}{\beta} \sum_{h'=1}^{h} H(d_{h'}, d^t_{h'}) \right),$$

$$\text{s.t.} \quad E^T d_h = F^T d_{h-1} \quad \forall h \in [H]. \tag{7}$$

**Theorem 10.** *The policy $\pi^{t+1}_h$ with occupancy measure $d^{t+1}_h$ defined in Eq. (7) can be computed as follows*

$$\pi^{t+1}_h(a|s) = \frac{\pi^t_h(a|s) \exp\left( \frac{\beta}{H-h+1} Q^t_h(s,a) \right)}{\sum_{a' \in \mathcal{A}} \pi^t_h(a'|s) \exp\left( \frac{\beta}{H-h+1} Q^t_h(s,a') \right)}, \qquad \forall s \in \mathcal{S}, \ \forall a \in \mathcal{A}.$$

*where $Q^t_h$ and $V^t_{h+1}$ satisfies the following recursion*

$$Q^t_h = \tilde{r}_h + F V^t_{h+1}$$
$$V^t_{h+1}(s) = \frac{H-h+1}{\beta} \log \sum_a \pi^t_h(a|s) \exp\left( \frac{\beta}{H-h+1} Q^t_{h+1}(s,a) \right).$$

**Remark 4.** *The above recurrencies are sometimes called soft Bellman equations Ziebart (2010); Fox et al. (2015).*

*Proof.* Let us introduce an auxiliary variable $\mu_h = d_h$ for all $h \in [H]$, then we can rewrite the optimization program as

$$\underset{d \in \Delta^H}{\arg\max} \ \underset{\mu}{\max} \sum_{h=1}^{H} \left( \langle \mu_h, \tilde{r}_h \rangle - \frac{1}{\beta} \sum_{h'=1}^{h} H(d_{h'}, d^t_{h'}) \right)$$

$$\text{s.t.} \quad E^T d_h = F^T \mu_{h-1} \quad \forall h \in [H]$$

$$\text{s.t.} \quad \mu_h = d_h \quad \forall h \in [H].$$

Notice that importantly, we do not constrain the variable $\mu$. Then, by Lagrangian duality we have that

$$\underset{d \in \Delta^H}{\max} \ \underset{\mu}{\max} \ \underset{Q,V}{\min} \sum_{h=1}^{H} \langle \mu_h, \tilde{r}_h \rangle - \frac{1}{\beta} \sum_{h'=1}^{h} H(d_{h'}, d^t_{h'})$$

$$+ \left\langle -E^T d_h + F^T \mu_{h-1}, V_h \right\rangle + \langle Q_h, d_h - \mu_h \rangle$$

$$= \underset{d \in \Delta^H}{\max} \ \underset{\mu}{\max} \ \underset{Q,V}{\min} \sum_{h=1}^{H} \langle \mu_h, \tilde{r}_h + F V_{h+1} - Q_h \rangle + \langle d_h, Q_h - E V_h \rangle$$

$$- \frac{1}{\beta} \sum_{h'=1}^{h} H(d_{h'}, d^t_{h'}) + \langle \nu_1, V_1 \rangle$$

$$= \underset{Q,V}{\min} \ \underset{d \in \Delta^H}{\max} \ \underset{\mu}{\max} \sum_{h=1}^{H} \langle \mu_h, \tilde{r}_h + F V_{h+1} - Q_h \rangle + \langle d_h, Q_h - E V_h \rangle$$

$$- \frac{H-h+1}{\beta} H(d_h, d^t_h) + \langle \nu_1, V_1 \rangle = \tilde{\mathcal{L}}^\star,$$

---

[7]The sum over previous stages is taken to ensure 1-strong convexity. Indeed, it holds that $\sum_{h'=1}^{h} H(d_{h'}, d'_{h'}) \geq D(d_h, d'_h) \geq \frac{1}{2} \|d_h - d'_h\|_1^2$. The first inequality is proven in Neu & Olkhovskaya (2021, Lemma 7).

where the last equality holds by Lagrangian duality and by $\sum_{h=1}^{H} \sum_{h'=1}^{h} H(d_{h'}, d_{h'}^t) = \sum_{h=1}^{H} (H - h + 1) H(d_{h'}, d_{h'}^t)$. Now since $\mu$ is unconstrained we have that $\max_{\mu} \sum_{h=1}^{H} \langle \mu_h, \tilde{r}_h + FV_{h+1} - Q_h \rangle$ is equivalent to impose the constraint $\tilde{r}_h + FV_{h+1} = Q_h$ for all $h \in [H]$. Moreover, as in the proof of Thm. 9 the optimal $d_h$ needs to satisfies that $\pi_{d_h}(a|s) = d_h(s,a)/\sum_a d_h(s,a)$ is equal to $\pi_h^+(Q, V; s) = \pi_h^t(\cdot|s) \odot \exp\left(\frac{\beta}{H-h+1}(Q_h(s,\cdot) - V_h(s))\right)$ for $V_h(s) = \frac{H-h+1}{\beta} \log \sum_a \pi_h^t(a|s) \exp(\frac{\beta}{H-h+1}Q_h(s,a))$. Plugging in, these facts in the expression for $\tilde{\mathcal{L}}^\star$, we have that

$$\tilde{\mathcal{L}}^\star = \min_Q \langle \nu_1, V_1 \rangle \quad \text{s.t.} \quad \tilde{r}_h + FV_{h+1} = Q_h \quad \forall h \in [H].$$

Since the above problem as only one feasible point, we have that the solution is the sequence $Q_h^t$ satisfying the recursion $\tilde{r}_h + FV_{h+1}^t = Q_h^t$ with $V_h^t(s) = \frac{H-h+1}{\beta} \log \sum_a \pi_h^t(a|s) \exp(\frac{\beta}{H-h+1}Q_h^t(s,a))$. $\qquad\square$

### E.3 Approximating soft Bellman equations by standard Bellman equations.

Unfortunately, implementing the update for the $V$ value as in Theorem 9 is often numerically instable. In this section, we show a practical approximation which is easy to implement and shown to be accurate for $\beta$ sufficiently small. In particular, we prove here Thm. 7.

### E.4 Proof of Thm. 7

*Proof.*

$$\frac{1}{\beta_h} \log \sum_a \pi_h^t(a|s) \exp(\beta_h Q_h^t(s,a)) \geq \frac{1}{\beta_h} \sum_a \pi_h^t(a|s) \log \exp(\beta_h Q_h^t(s,a))$$
$$= \langle \pi_h^t(\cdot|s), Q_h^t(s,\cdot) \rangle,$$

where the above inequality holds for Jensen's. For the upper bound, we first use the inequality $e^x \leq 1 + x + x^2$ for $x \leq 1$ we have that

$$\frac{1}{\beta_h} \log \sum_a \pi_h^t \exp(\beta_h Q_h^t(s,a))$$
$$\leq \frac{1}{\beta_h} \log \sum_a \pi_h^t (1 + \beta_h Q_h^t(s,a) + \beta_h^2 Q_{\max}^2) \quad (\text{Using } Q_h^t(s,a) \leq Q_{\max})$$
$$= \frac{1}{\beta_h} \log(1 + \beta_h \sum_a \pi_h^t(a|s) Q_h^t(s,a) + \beta_h^2 Q_{\max}^2)$$
$$\leq \frac{1}{\beta_h} \left( \sum_a \pi_h^t(a|s) \beta_h Q_h^t(s,a) + \beta_h^2 Q_{\max}^2 \right) \quad (\text{Using } \log(1+x) \leq x)$$
$$\leq \langle \pi_h^t(\cdot|s), Q_h^t(s,\cdot) \rangle + \beta_h Q_{\max}^2.$$

$\qquad\square$

**Remark 5.** *Given this result, in the implementation for deep RL experiment, i.e. Algorithm 2 we compute the standard $Q$ value satisfying the standard Bellman equations (given in Lemma 2) rather than the soft Bellman equation in Thm. 9. In virtue of Thm. 7, the approximation is good for $\beta$ reasonably small.*

## F Supplementary material on experiment

### F.1 Experiment in MT-bench 101

The tasks in MT-bench 101 include Context Memory (CM), Anaphora Resolution (AR), Separate Input (SI), Topic Shift (TS), Content Confusion (CC), Content Rephrasing (CR), Format Rephrasing (FR), Self-correction (SC), Self-affirmation (SA), Mathematical Reasoning (MR), General Reasoning (GR), Instruction

Clarification (IC), and Proactive Interaction (PI). We list the description of each task in Tab. 5. The default evaluation mode of MT-bench 101 is that the GPT model requires to access the conversation based on the given ground truth of previous steps, provided in MT-bench 101. However, in our problem setting, the answers among the conversation is also generated by the model. We use "gpt-4o-mini-2024-07-18" to evaluate the conversation. The maximum output length and maximum sequence length of gpt-4o are set as 4096. We use a batch size of 8 with a temperature of 0.8. We use the same prompt for gpt-4o as in Bai et al. (2024). Our experiment is conducted on 4 H200 GPUs. We use the PyTorch platform and the Transformer Reinforcement Learning (TRL) for fine-tuning. The $\gamma$ is selected as zero. Each method is trained with epochs number selected from $\{1, 2\}$, learning rates from $\{5e\text{-}6, 5e\text{-}7\}$, and $\beta$ values from $\{0.1, 0.01, 0.001\}$. The final model is chosen based on the highest winning rate against the base model, as determined by the PairRM model. We use full-parameter fine-tuning for all methods with bf16 precision. A batch size of 64 is used. The maximum output length and maximum prompt length during training are both set as 2048. We use AdamW optimizer (Loshchilov & Hutter, 2019) and cosine learning rate schedule (Loshchilov & Hutter, 2017) with a warmup ratio of 0.1.

Table 5: A detailed description of each task in MT-bench 101 (taken from Bai et al. (2024).)

| Task | Abbr. | Description |
|---|---|---|
| Context Memory | CM | Recall early dialogue details to address the user's current question. |
| Anaphora Resolution | AR | Identify pronoun referents throughout a multi-turn dialogue. |
| Separate Input | SI | The first turn outlines the task requirements and the following turns specify the task input. |
| Topic Shift | TS | Recognize and focus on the new topic when users unpredictably switch topics. |
| Content Confusion | CC | Avoid interference from similar-looking queries with distinct meanings in the dialogue's history. |
| Content Rephrasing | CR | Rephrase the content of the last response according to the user's newest requirement. |
| Format Rephrasing | FR | Rephrase the format of the last response according to the user's newest requirement. |
| Self-correction | SC | Recorrect the last response according to the user feedback. |
| Self-affirmation | SA | Preserve the last response against inaccurate user feedback. |
| Mathematical Reasoning | MR | Collaboratively solve complex mathematical problems with users across dialogue turns. |
| General Reasoning | GR | Collaboratively solve complex general reasoning problems with users across dialogue turns. |
| Instruction Clarification | IC | Seek clarification by asking further questions on ambiguous user queries. |
| Proactive Interaction | PI | Propose questions in reaction to user statements to spark their interest to continue the dialogue. |

Next, we provide the comparison between the proposed MPO and IPO (Azar et al., 2024), which also uses the squared loss and bypasses the BT model assumption. We run both IPO and MPO for one iteration. The results in Tab. 6 show that MPO achieves a higher average score than IPO.

Table 6: Comparison between MPO and IPO in MT-BENCH 101 dataset.

| Model | Avg. | Perceptivity | | | | | Adaptability | | | | | | Interactivity | |
|---|---|---|---|---|---|---|---|---|---|---|---|---|---|---|
| | | CM | SI | AR | TS | CC | CR | FR | SC | SA | MR | GR | IC | PI |
| Base (Mistral-7B-Instruct) | 6.223 | 7.202 | 7.141 | 7.477 | 7.839 | 8.294 | 6.526 | 6.480 | 4.123 | 4.836 | 4.455 | 5.061 | 5.818 | 5.641 |
| IPO | 6.498 | 7.518 | 7.480 | 7.759 | 7.952 | 8.652 | 6.892 | 6.768 | 4.390 | 5.185 | 4.313 | 5.378 | 6.146 | 6.044 |
| MPO | 6.630 | 7.624 | 7.846 | 8.085 | 8.398 | 8.947 | 7.105 | 7.286 | 4.208 | 4.993 | 4.377 | 5.264 | 6.179 | 5.873 |

We now present an ablation study to evaluate the benefits of incorporating terminal rewards. Using MPO, we compare two approaches for optimizing $a_h$: one computes the preference signal based on the terminal state $s_{H+1}$, while the other uses the immediate next state $s_h$. The results within one iteration for the MT-Bench 101 dataset are shown in Tab. 8, and those for the GSM/Math experiments are provided in Tab. 7. Our findings reveal that using the terminal state $s_{H+1}$ performs worse than using the immediate state $s_h$ in MT-Bench 101. In contrast, the difference in performance is negligible in the GSM/Math tasks. The

underlying reason is that in multi-turn conversational datasets, especially when adjacent questions are not closely related, relying on preferences derived from the terminal state can introduce noise. However, in math and reasoning tasks, the terminal state often captures the final answer, making it more critical. Moreover, using $s_{H+1}$ for preference signals is significantly more computationally expensive than using $s_h$, due to the extended sequence length. Consequently, we conclude that adapting the choice of terminal preference or intermediate preference on the task's characteristics is crucial for balancing performance and efficiency.

### F.2 Experiment in math-reasoning task

Our experiment is conducted on 4 A100 GPUs. For both `MPO` and `OMPO`, we perform full-parameter finetuning for 1 epoch with learning rate $5e^{-7}$ and $\beta$ tuned in the range of $\{0.1, 0.01, 0.001\}$, we set the $\log z$ as 0.5. The final state with the answer is important in this task so we only use the terminal reward (see Tab. 7 for comparison). We run two iterations for both methods. We use AdamW optimizer (Loshchilov & Hutter, 2019) and cosine learning rate schedule (Loshchilov & Hutter, 2017) with a warmup ratio of 0.1.

Table 7: Ablation on terminal reward in MATH and GSM8K dataset.

| Method | GSM8K | Math |
|---|---|---|
| Base (Qwen2-7B-Instruct) | 0.8559 | 0.5538 |
| MPO (intermediate reward) | 0.8734 | 0.5720 |
| MPO (terminal reward) | 0.8734 | 0.5734 |

### F.3 Additional ablation study and experimental detail

We conduct an ablation study on the math reasoning task for OMPO at the second iteration, using various combinations of $(\beta, \log z)$. The method generally performs robustly across a wide range of parameter settings, except for the case of $(\beta, \log z) = (0.001, 0.5)$, which shows noticeably degraded performance. On the other hand, we adopt a learning rate of $5 \times 10^{-7}$ following the StepDPO paper (Lai et al., 2024), as we observed that using larger or smaller learning rates often causes training to fail to reach sufficiently low loss values (lower than 0.5), similar to the failure mode observed for $(\beta, \log z) = (0.001, 0.5)$.

Table 9: Comparison of GSM and Math results for various $(\beta, \log z)$ settings.

| $(\beta, \log z)$ | GSM | Math |
|---|---|---|
| (0.1, 0) | 0.8711 | 0.5730 |
| (0.01, 0) | 0.8741 | 0.5762 |
| (0.001, 0) | 0.8810 | 0.5774 |
| (0.1, 0.5) | 0.8779 | 0.5786 |
| (0.01, 0.5) | 0.8711 | 0.5774 |
| (0.001, 0.5) | 0.1713 | 0.1396 |

Regarding the experimental details, as also mentioned in the previous subsection, we use 4 H200 GPUs for the MT-Bench 101 experiment and 4 A100 GPUs for the math reasoning experiment. The training time per run is approximately one hour and two hour for the MT-Bench 101, and math reasoning task, respectively. The total time scales with the number of hyperparameter searches, as mentioned earlier. Also the total time scales with the numer of algorithms' iterations, where we use OMPO(iter=3). The evaluation time on MT-Bench 101 depends on the network, as we use GPT-4o-mini for evaluation, which generally takes around 30 minutes. For the math reasoning task, we completely follow the training and evaluation pipeline from Lai et al. (2024). In their paper, they provided a math reasoning dataset as a training set, which includes

Table 8: Ablation on terminal reward in MT-BENCH 101 dataset.

| Model | Avg. | Perceptivity | | | | | Adaptability | | | | | | Interactivity | |
| | | CM | SI | AR | TS | CC | CR | FR | SC | SA | MR | GR | IC | PI |
|---|---|---|---|---|---|---|---|---|---|---|---|---|---|---|
| Base (Mistral-7B-Instruct) | 6.223 | 7.202 | 7.141 | 7.477 | 7.839 | 8.294 | 6.526 | 6.480 | 4.123 | 4.836 | 4.455 | 5.061 | 5.818 | 5.641 |
| MPO (intermediate reward) | 6.630 | 7.624 | 7.846 | 8.085 | 8.398 | 8.947 | 7.105 | 7.286 | 4.208 | 4.993 | 4.377 | 5.264 | 6.179 | 5.873 |
| MPO (terminal reward) | 6.459 | 7.536 | 7.328 | 7.643 | 8.084 | 8.518 | 6.847 | 6.883 | 4.357 | 4.863 | 4.403 | 5.542 | 6.034 | 5.924 |

10,795, which is a mixture from the GSM8k and the Math dataset. For the testing set, In GSM dataset, there are 1319 questions while in the math dataset there are 5000 questions, and the data can be bound in StepDPO's repository [8]. For the MT-bench-101, we use the dataset provided in [9]. The MT-Bench-101 dataset contains 4208 turns across 13 tasks, as described in Bai et al. (2024). Due to limited dataset size and the fact that MT-Bench-101 does not provide any ground-truth preference labels, we use its prompts as the source of conversations during training, while the actual supervision comes exclusively from the Pairwise Reward Model (PairRM). Thus, the model does not memorize human preference scores or annotations from MT-Bench-101. For evaluation, we follow the standard practice in iterative preference optimization and use a separate judge model (GPT-4o-mini via the OpenCompass platform) to score conversations generated by our trained models. The evaluation judge is completely independent of the PairRM used for training. We conduct inference with the following settings: 16 queries per second, maximum output length of 4096 tokens, maximum sequence length of 4096 tokens, batch size of 8, and sampling temperature 0.8. The resulting scores are competitive with those reported in Bai et al. (2024), with the difference being that we use GPT-4o-mini as the judge instead of GPT-4o.

## G    Discussion on the Eq. (Game) objective

In this section, we elaborate on the Eq. (Game) objective for multi-step alignment.

**Discussion on the** $\arg\max\min$. By $\arg\max_\pi min_{\pi'}$, we refer to getting the saddle point of the problem, so that a policy pair is returned. The considered game has antisymmetry property of the preference relation, i.e., $\mathbb{P}(y \succ y') = 1 - \mathbb{P}(y \prec y')$. This antisymmetry implies that if $(\pi^\star, \hat{\pi}^\star)$ is a Nash equilibrium (NE), then so is $(\hat{\pi}^\star, \pi^\star)$. Moreover, by the interchangeability of NE strategies in two-player constant-sum games, $(\pi^\star, \pi^\star)$ and $(\hat{\pi}^\star, \hat{\pi}^\star)$ must also be NE (Nash, 1951). Therefore, the optimal policies coincide.

**Different prompts** $x$ **and different horizon** $H$**.** In the notation section, the preference between two sentences $[x, a]$ and $[x', a']$ is defined as $\mathbb{P}([x, a] \succ [x', a'])$ as a general definition.

Considering the special case $H = 1$, the objective reduces to $(\pi^*, \pi^*) = \arg\max_\pi \min_{\pi'} \mathbb{E}_{x_1, a_h, a'_h} \mathbb{P}([x_1, a_1] \succ [x_1, a'_1])$. Therefore, there is no need to consider preference on different $x$.

Considering $H > 1$, we need to calculate $\mathbb{P}([s_h, a_h] \succ [s'_h, a'_h])$, note that $s_h = [s_{h-1}, a_{h-1}, x_h]$, $s'_h = [s'_{h-1}, a'_{h-1}, x_h]$ where $x_h$ is the same question in the $h$ step for both player in multi-turn conversation tasks, or empty in multi-step reasoning tasks. Therefore, the comparison still does not involve two completely unrelated questions, contrary to the reviewer's example.

In our experimental datasets MT-Bench101 and GSM8k/Math, $s_h$ and $s'_h$ are highly correlated within the same topic. This makes it reasonable for the reward model to score based on the current state output. However, we agree that mitigating the effect of previous answers or adding penalties for earlier steps could be valuable directions for future work when designing the reward model.

---

[8]https://github.com/dvlab-research/Step-DPO/tree/main/data/test
[9]https://github.com/mtbench101/mt-bench-101

The horizon $H$ in the objective can be taken as the maximum horizon among all questions. So even if problem A has shorter trajectory (e.g., 2 step) compared to B (e.g., 3 steps), we have $\mathbb{P}([s_h, a_h] \succ [s'_h, a'_h]) = 1/2$ for step $h = 3$ where both players have empty answers. Therefore, the constant sum is 3, which is the same for problems A, B.

Regarding the steps in the reasoning dataset, in theory, it corresponds to the maximum horizon across all questions as discussed above. In the practical implementation of our algorithm, at each step, the model generates different answers and performs preference optimization. Therefore, the number of steps for each question is determined by the model itself. Once the model outputs the final answer, the process ends.

**Minimal example for the benefit of general preference $\mathbb{P}$.** The BT assumption implies transitivity. This is restrictive because the preference dataset collected from different humans might not be transitive even if each human follows a transitive model in generating the preference. As an example, consider 3 humans $e_1, e_2, e_3$ and 3 answers $y_1, y_2, y_3$, denote $o_e$ the preference model of human $e$. They follow these preferences:

$$o_{e_1}(y_1 \succ y_2) = 1, \quad o_{e_1}(y_2 > y_3) = 0, \quad o_{e_1}(y_3 \succ y_1) = 1.$$
$$o_{e_2}(y_1 \succ y_2) = 0, \quad o_{e_2}(y_2 \succ y_3) = 1, \quad o_{e_2}(y_3 \succ y_1) = 1.$$
$$o_{e_3}(y_1 \succ y_2) = 1, \quad o_{e_3}(y_2 \succ y_3) = 1, \quad o_{e_3}(y_3 \succ y_1) = 0.$$

Each of these models is transitive. However, the average preference model defined as $\mathbb{P}(y \succ y') = \frac{1}{3}\sum_{e \in \{e_1, e_2, e_3\}} o_e(y \succ y')$ satisfies $\mathbb{P}(y_1 \succ y_2) = \mathbb{P}(y_2 \succ y_3) = \mathbb{P}(y_3 \succ y_1) = 2/3$. Thus, the average model is non transitive and can not be modeled by the BT assumption. Therefore, the BT assumption is data wasteful. In this example, one should consider preferences only from a single human in order to make the BT assumption valid. Not enforcing the BT assumption allows the use of more data, i.e., preferences from all three humans. Thus, DPO is developed based on the assumption of BT model, which can not capture such intransitive preference. Moreover, the Nash Equilibrium (NE) policy $\pi^\star$ guarantees a win rate greater than 50% against any other policy. This follows by the definition of NE: $\mathbb{P}(\pi^\star \succ \pi) \geq \mathbb{P}(\pi^\star \succ \pi^\star) = 50\%$ for any $\pi$.

**Minimal example for the benefit of intermediate reward.** In multi-turn conversation tasks, such as MT-bench 101 (Bai et al., 2024), the user asks questions $x_1$, $x_2$, $x_3$, and receives answers $a_1$, $a_2$, $a_3$. When $x_2$ is not closely related to $x_1$, aligning the first step using feedback among different $a_1$ is much more helpful than using the sequence $[a_1, x_2, a_2]$, where $x_2, a_2$ can be considered as noise. In mathematical reasoning tasks, as mentioned in Lai et al. (2024), some cases yield correct final answers but contain errors in intermediate reasoning steps. Consequently, Lai et al. (2024) filter out such samples using GPT-4. For example, consider a case where the reasoning steps yield a correct final answer but include an error: $[a_1^{\text{correct}}, a_2^{\text{wrong}}, a_3^{\text{correct}}]$, where $a_2^{\text{wrong}}$ is incorrect while all of the other steps and the final answer $a_3^{\text{correct}}$ is correct. When there is another response, $[a_1^{\text{correct}}, a_2^{\text{correct}}, a_3^{\text{correct}}]$ with all correct steps, using only terminal signal for aligning step 2 might not guarantee that $a_2^{\text{correct}} \succ a_2^{\text{wrong}}$ because both of final answers are correct, especially when there is only an incorrect step among long reasoning steps. In contrast, an intermediate signal would clearly indicate $a_2^{\text{correct}} \succ a_2^{\text{wrong}}$, accurately reflecting the quality of the intermediate steps. In practice, if the final signal is important, e.g., in math reasoning task, then we can use only the terminal reward or the average of terminal reward and intermediate reward, otherwise one can just use the intermediate reward, which is cheaper to collect as compared to assigning reward until the terminal state.

**Availability of preference oracle $\mathbb{P}$.** Online preference signals are ideally obtained from human annotators while it is prohibitively expensive in practice, limited by human capability, and often beyond the reach of the open-source community (Dong et al., 2024). Prior work has demonstrated that training a preference reward model (RM) and using it to generate labels in a semi-supervised fashion can significantly boost model performance (Wu et al., 2025; Tran et al., 2023; Sessa et al., 2025). Notably, (Tran et al., 2023) shows that the 0.4B Pair-RM (used in Sec. 4) can support iterative preference learning and get strong performance on AlpacaEval-2. Process-reward-models are gaining significant attention due to their reliable inference-time scaling (Zhang et al., 2025b;a). The llama-3-RM used in our paper is also trained on a multi-turn dataset. More recently, process-based self-rewarding language models (Zhang et al., 2025b) are introduced to integrate the reward model and the policy into a single model. We believe that as LLMs continue to improve, they can increasingly serve as their own evaluators—following the "LLM-as-a-judge" paradigm (Zheng et al., 2023;

Zhang et al., 2025b) and autoregressive RM (Xu et al., 2025). This makes it reliable to automate per-step feedback using LLM itself rather than humans.

