# OpenReview forum: "Multi-Step Alignment as Markov Games: An Optimistic Online Mirror Descent Approach with Convergence Guarantees"
_TMLR — Accepted by TMLR_

### Review · Reviewer_CxpN · 2025-10-26

**Summary Of Contributions:**

- They propose an optimistic online mirror descent algorithm called OPMO for multi-turn preference optimization that operates in the occupancy measure space
- They show provable convergence guarantees that does not depend on a regularization strength.
- They reformulate the occupancy update step in their algorithm with policy and value function updates using soft-Bellman equations for a more efficient implementation
- On a simple tabular experiment, their OPMO converges 20x faster than a non-optimistic version of their algorithm
- Averaged across tasks in MT-Bench-101, a multi-turn conversation dataset, OPMO outperforms MPO, Step-DPO, SPPO, and DPO
- On alignment datasets of MATH and GSM8K, OPMO outperforms baselines on the average of the two datasets.

**Audience:**

Yes

**Audience Explanation:**

Since human preferences can be non-transitive, it is interesting that the proposed method bypasses the Bradley-Terry assumption that is not often questioned when using DPO-like methods. Using an intermediate preference signal is desirable because preferences might vary over the course of a multi-turn conversation, which often occurs in user-application settings. Provable convergence guarantees are interesting because collecting preference data in multi-turn conversation settings can be expensive.

**Broader Impact Concerns:**

None.

**Claims And Evidence:**

Yes

**Claims Explanation:**

- The authors claim that their method aligns preferences at every turn of a multi-turn conversation compared to other methods that are only either one-turn or align preferences at the terminal state. This is achieved by the objective (Game) that is multi-turn and is based on intermediate rewards.
- The authors claim that their method is able to model non-transitive preferences which it does because the reward function comes directly from the preference oracle
- The authors claim that their method has robust convergence guarantees that also does not deteriorate for a vanishing regularization strength. They show this by proving Theorem 4, that gives the number of policy updates necessary for their algorithm to output a policy that is an epsilon-approximate Nash equilibrium of the two-player game.
- They demonstrate that on multi-turn conversation tasks and mathematical reasoning tasks, their approach outperforms relevant baselines such as Step-DPO and MPO, validating the claim that their method is effective.

**Requested Changes:**

The following two changes are not critical for securing my recommendation:
- First equation in 2.3: Q is missing expectation
- Table 2 is missing values for OPMO(iter=1)

---

> ### Author Response · Authors · 2025-11-24
> **Response to reviewer CxpN**
>
> We thank reviewer CxpN for the thoughtful feedback, and the revisions are highlighted in blue in the revised version.
>
> > 1. Equ 2.3: Q is missing expectation
>
> We fixed the issue in the updated version.
>
> > 2. Table 2 is missing values for OPMO(iter=1)
>
> For OMPO, we omit the result for iteration 1, because it is the same as MPO. OMPO relies on information from two adjacent steps, and at the initial step there is no previous step, so it coincides with MPO. We have added a sentence in the caption of Table 2.

---

### Review · Reviewer_sWT8 · 2025-10-28

**Summary Of Contributions:**

The paper proposes OMPO for multi-step preference alignment via a two-player constant-sum Markov game, with theoretical guarantees and promising experiments. In particular, the authors have provided the following contributions:

1, A Markov-game formulation enabling intermediate-step alignment with general (non-transitive) preferences.

2, An algorithm named OMPO: optimistic mirror-descent style updates with claimed $O(1/\varepsilon)$ iteration complexity.

3, A projection-free policy update via a dual formulation and soft Bellman equations connecting occupancy updates to policies.

4, Empirical evidence of improved multi-turn dialogue quality and competitive math performance without heavy supervision.

**Audience:**

Yes

**Audience Explanation:**

Yes. The findings will interest TMLR’s audience because multi-step alignment, optimistic mirror descent, and game-theoretic formulations of RLHF are currently active research areas in both machine learning theory and LLM alignment. The proposed OMPO framework contributes to bridging theoretical guarantees and practical RLHF methods.

**Broader Impact Concerns:**

No.

**Claims And Evidence:**

Yes

**Claims Explanation:**

Yes. The theoretical claims (occupancy-measure formulation, convergence of OMPO, and the projection-free dual implementation) are supported by mathematically consistent proofs, with only minor missing things that can be fixed by clarification. The empirical evidence—improved MT-Bench dialogue scores and competitive math-reasoning accuracy—convincingly demonstrates the method’s effectiveness. Overall, the paper provides clear and verifiable support for its main claims.

**Requested Changes:**

p.6, theorem 4: In the statement and proof of Theorem 4, we did not find an explicit global reward-range assumption (e.g., $r \in [0,1]$ or $r \in [-1,1]$), nor a stated dual-norm gradient bound $G$ required for the optimistic mirror descent analysis. Likewise, the stepsize condition $\beta$ is stated but not connected to $G$ and the strong convexity parameter $\lambda$, as is standard in OMD proofs. Could you please (i) clarify the reward range assumed in the analysis, (ii) derive the corresponding gradient bound $G$ under your chosen norm pairing, and (iii) state the precise stepsize constraint (e.g., $\beta \le c/(G\sqrt{\lambda})$, or your equivalent) and indicate where it is invoked in the proof?

p.6, theorem 5: The addition of the constant term $\langle \phi, \mathbf{1} \rangle$ does not affect the optimizer because all feasible occupancy measures in $\mathcal{F}_{s_1}$ have fixed total mass. Could you add a brief remark clarifying this invariance, so that the one-player equivalence argument is fully transparent to the reader?

p.6, theorem 6: The last-iterate argument appears to assume an interior equilibrium. Would you add a sentence explaining behavior when the Nash equilibrium lies on the boundary (common in zero-sum) and whether entropic smoothing (or similar) is required?

p.8-10, The experimental section does not report stability observations for the optimistic updates, nor does it clearly state the learning rates used, the value of $\gamma$ (it appears that $\gamma=0$ may have been used), or any sensitivity analysis. Could you report whether any training oscillations were observed, list the learning rates and $\gamma$ used in each experiment, and briefly discuss the sensitivity of OMPO to these hyperparameters?

p.8-10, Would you detail the baseline settings (or make more comments) to ensure fairness and reproducibility, such as precise settings for $K$ (number of sampled conversations), best/worst selection, evaluator configuration, number of updates, and compute parity with OMPO?

p.19, Can you clarify what do you mean by ``the only initial state that can lead to s''? Is this an assumption or can be explicitly proven? What if multiple initial states could reach s?

---

> ### Author Response · Authors · 2025-11-24
> **Response to reviewer sWT8**
>
> We thank reviewer sWT8 for the thoughtful feedback, and the revisions are highlighted in blue in the revised version.
>
> > 1. theorem 4: In the statement and proof of Theorem 4, we did not find an explicit global reward-range assumption (e.g., or ), nor a stated dual-norm gradient bound required for the optimistic mirror descent analysis
>
> Thanks for this comment, as present at page 3, the reward function is defined as $r(x,a,x',a') := \mathbb{P}([x,a] \succ [x',a'])$. Therefore, we have that $\| r \|_{\infty}\leq 1$.
> Such reward norm bound is exactly the dual-norm gradient bound needed in the analysis of Optimistic Mirror Descent Ascent. We have added this comment in the main text.
>
>  > 2. theorem 5: The addition of the constant term does not affect the optimizer because all feasible occupancy measures in  have fixed total mass. Could you add a brief remark clarifying this invariance, so that the one-player equivalence argument is fully transparent to the reader?
>
>  Thanks for the suggestion! We have added a remark on that after the statement of theorem 5.
>  > 3. theorem 6: The last-iterate argument appears to assume an interior equilibrium. Would you add a sentence explaining behavior when the Nash equilibrium lies on the boundary (common in zero-sum) and whether entropic smoothing (or similar) is required?
>
> Thanks a lot for the comment, we have added a remark after Theorem 6.
>
> > 4. experimental section does not report stability observations for the optimistic updates, nor does it clearly state the learning rates used.
>
> In our original version of the submission. we have provided detail on the experimental setup in the appendix. To make this more clear, in the revisied version, we added an additional sentence in the main body of the experiment to highlight this, i.e., we add "Additional description of the dataset and setup, and detail on the hardware setup, hyperparameter setup, evaluation setup, and ablation studies can be found in Appendix F". Additionally, we provide additional ablation study in the appendix for the value of $\beta$ and $log Z$.
>
>
> > 5.  Detail on Baseline settings.
>
> Regarding to the baseline setting, We follow the source code of StepDPO, SPPO, and we perform the same evalaution configuation. We perform the same searching of the learning rate as well as the beta, as stated in the appendix of the original submission. For instance, we have the following paragraph in our original submission
>
> The maximum output length and maximum sequence length of gpt-4o are set as 4096. We use a batch size of 8 with a temperature of 0.8. We use the same prompt for gpt-4o as in \citet{bai2024mt}. Our experiment is conducted on 4 H200 GPUs. We use the PyTorch platform and the Transformer Reinforcement Learning (TRL) for fine-tuning. The $\gamma$ is selected as zero. Each method is trained with epochs number selected from $ \{1, 2\} $, learning rates from $\{5e-6, 5e-7\} $, and $\beta$ values from $ \{0.1, 0.01, 0.001\}$. The final model is chosen based on the highest winning rate against the base model, as determined by the PairRM model. We use full-parameter fine-tuning for all methods with bf16 precision. A batch size of 64 is used. The maximum output length and maximum prompt length during training are both set as 2048.
>
> For the math reasoning task, we found that the learning rate $5.0e^{-7}$ provided in StepDPO paper~\citet{bai2024mt} bring stable training without diverging loss or collapsed policies for other method as well so we adopt it. Systematic ablation and quantify the effect of parameters can be found at appendix F.
>
>
>
> > 6. What do you mean by ``the only initial state that can lead to s''? Is this an assumption or can be explicitly proven? What if multiple initial states could reach $s$?
>
> Given the definition of the dynamics in section 2.2 it can not happen that a state can be reached from different initial states. This is a characterist of the token level MDP used in LLMs modeling. We have clarified this in Section 2.2.

---

> > ### Comment · Reviewer_sWT8 · 2025-11-30
> > **No further questions**
> >
> > Thank you for your response and revisions. My concerns have been addressed, and I have no further questions.

---

### Review · Reviewer_nqhM · 2025-11-12

**Summary Of Contributions:**

### **Summary of Contributions**

1. The paper reformulates multi-step preference alignment as a two-player zero-sum Markov game in the occupancy measure space, converting the non-convex objective into a bilinear saddle-point problem.

2.  Building on this formulation, the authors introduce the OMPO algorithm, which provably reaches an ε-Nash equilibrium within O(ε⁻¹) iterations under standard convexity and strong convexity assumptions.

3. A projection-free implementation is proposed by replacing soft Bellman updates with standard Q-learning approximations and regressing the partition constant, enabling computational efficiency for large-scale LLM alignment.

4. Experiments on MT-bench-101 and GSM8K/MATH show that OMPO and its variant consistently outperform or match strong baselines such as DPO, Step-DPO, and SPPO, without requiring additional supervision like step-level error annotations.

---

### **Strengths**
1. Clear and rigorous theoretical foundation linking multi-step preference alignment with occupancy-measure optimization.

2. Provable convergence guarantees combined with a practical, implementable algorithm.

3.  Consistent empirical improvements across dialogue and reasoning benchmarks.

---

### **Weaknesses**

1. The convergence analysis relies on trajectory independence between agents and a finite-horizon assumption. These simplifications make the proofs tractable but limit the applicability to real multi-turn dialogue scenarios where one agent’s action affects the other’s state and response.

2. The theory establishes only asymptotic convergence through quasi-Féjer sequences, without explicit finite-sample or last-iterate bounds. This leaves uncertainty about empirical convergence speed, stability, and efficiency in large-scale alignment.

3. The practical OMPO implementation uses heuristic relaxations, such as small-β approximations for soft Bellman equations and regression-based partition constant estimation (log Z). These approximations are not analyzed for bias or variance and could affect stability or accuracy.

4. Model selection and evaluation both depend on LLM-based judges. This shared bias may inflate results, as there is no cross-judge or human evaluation to verify alignment quality independently.

6. Experiments focus only on MT-bench-101 and math reasoning datasets, without testing on other domains or modalities. Details about data splits, training cost, and random seed variance are missing.

**Audience:**

Yes

**Audience Explanation:**

The paper addresses multi-step preference alignment through a theoretically grounded occupancy-measure formulation and optimistic mirror optimization, which would interest researchers in reinforcement learning, large language model alignment, and optimization theory.

**Broader Impact Concerns:**

The paper’s broader impact discussion is generally adequate, as it focuses on methodological contributions without using human data.

**Claims And Evidence:**

Yes

**Claims Explanation:**

The paper’s theoretical claims are well supported within the stated assumptions, but these assumptions are restrictive and limit generality.
Empirical evidence shows promising results.

**Requested Changes:**

1. Provide a clearer discussion of the trajectory-independence and finite-horizon assumptions in the theoretical framework, including when they hold and how violations affect convergence.

2. Add finite-sample or last-iterate convergence analysis, or at least provide empirical evidence (e.g., convergence curves) to demonstrate practical stability.

3. Include systematic ablation and sensitivity studies on β, horizon H, temperature constants, and replay buffer size to validate robustness of the reported performance.

4. Expand the experimental scope to include other reasoning or dialogue benchmarks, or cross-domain tests (e.g., instruction-following or multimodal tasks).

5. Clarify implementation details such as dataset splits, training time, variance across random seeds, and hardware configuration to improve reproducibility.

7. Quantify the effect of the small-β approximation and log Z regression used in the practical algorithm; include either theoretical bounds or empirical diagnostics on these approximations.

8. Improve the clarity of algorithmic exposition—e.g., explicitly describe how OMPO transitions from the theoretical to practical form and where the approximations enter.

9. Add a short discussion on potential extensions to infinite-horizon or multi-agent general-sum settings.

---

> ### Author Response · Authors · 2025-11-24
> **Response to reviewer nqhM**
>
> We thank reviewer nqhM for the thoughtful feedback, and the revisions are highlighted in blue in the revised version.
>
> > 1. Clearer discussion of the trajectory-independence and finite-horizon assumptions in the theoretical framework, including when they hold and how violations affect convergence.
>
> The trajectory independence holds whenever the two responses to be compared are generated with two independent queries to the LLM. The finite horizon assumption is also quite natural and applies when a fixed maximum chat length is enforced and such limit is known in advance.
>
> > 2. Finite-sample or last-iterate convergence analysis.
>
> The contributions of the theoretical results and the comparisons to other work are presented in Table 1 for the reader. The result on the last-iterate asymptotic convergence analysis is provided in Theorem 6. While we only provide asymptotic result, according to Table 1, we believe that extending the anayisis of OMPO to last iteration under additional $\alpha$-strong convexity might be interesting as a future work where an additional KL term will be added into the objective.
>
>
> > 3. Systematic ablation and quantify the effect of the β log Z.
>
> We provided additional ablation in appendix F for the effect of $\beta$ and $Z$, and we can see that the method generally performs robustly across a wide range of parameter settings, except for the case of $(0.001, 0.5)$, which shows noticeably degraded performance.
>
>
> > 4 Expand the experimental scope to include other reasoning or dialogue benchmarks.
>
>
> We appreciate the reviewer’s suggestion to expand the experimental scope to include additional benchmarks. Due to limited time and computational resources, we were unfortunately unable to incorporate further benchmarks. In particular, the training with different hyperparameter search, pairwise preference scoring using the pair-RM model, and evaluation via GPT-based judges are all computationally expensive and time-consuming. For example, in the MT-Bench 101 task, we first run the reward model to score the generated outputs. Next, we perform a hyperparameter search over different learning rates $\{5\times10^{-6},5\times10^{-7}\}$ and $\beta$ values $\{0.1,0.01,0.001\}$, resulting in six experiments in total. Afterward, we use GPT-based evaluators to assess the final outputs. Note that this entire process corresponds to only one iteration of the OMPO algorithm, and we repeat it three times in total to obtain the final OMPO (iter=3) results.
>
> While we agree that including more benchmarks would strengthen the empirical validation, we would like to also **emphasize that this paper also contributes substantial theoretical analyses, including several  convergence guarantees and optimization insights as summarized in table 1**. Moreover, the current set of experiments—comprising one synthetic experiment, two benchmark studies, and several ablation analyses already provides strong and comprehensive evidence supporting the claims of the paper.
>
> > 5. Implementation details.
>
> In Appendix F of the original submission, we provided the experiment detail. To make this more clear, in the revised version, we added an additional sentence in the main body of the experiment to highlight this, i.e., we add "Additional description of the dataset and setup, and detail on the hardware setup, hyperparameter setup, evaluation setup, and ablation studies can be found in Appendix F".
>
> > 6 Improve the clarity of algorithmic exposition—e.g., explicitly describe how OMPO transitions from the theoretical to practical form and where the approximations enter.
>
> There are two approximations to implement OMPO in practice. First, as mentioned in the last paragraph at page 6, we assume that we are in the regime of small $\beta$ and we approximate the regularized value functions with unregularized value functions.
>
> Second, for computationally efficiency we approximate the online mirror ascent step with a regression problem as explained at the end of Section 3.
> > 7 Add a short discussion on potential extensions to infinite-horizon or multi-agent general-sum settings.
>
> We have added these as open directions in the conclusions section.

---

> > ### Comment · Reviewer_nqhM · 2025-12-03
> > **Response to the authors**
> >
> > Thank you for the revisions. The additional clarifications address most of my earlier comments, and I have no further concerns at this stage.

---

### Review · Reviewer_za9D · 2025-11-16

**Summary Of Contributions:**

**Summary of Contributions:**

This paper presents a novel LLM alignment framework that reformulates the problem from a single-step preference optimization (like DPO) into a two-player, constant-sum, partially-observable Markov game. The authors introduce the Optimistic Multi-step Preference Optimization (OMPO) algorithm, based on optimistic online mirror descent (OOMD), to find an optimal policy by maximizing a "win rate" at each step. The primary theoretical contribution is a rigorous proof that OMPO converges to an ϵ-approximate Nash equilibrium at a policy update rate of $O(\epsilon^{-1})$. This framework is empirically validated on multi-turn conversation and mathematical reasoning datasets.

**Strength:**

1. The authors provide formal, complete proofs for the main convergence theorems, establishing a strong foundation for the proposed OMPO algorithm and its $O(\epsilon^{-1})$ convergence rate.
2. The method demonstrates consistent and significant performance improvements over existing baselines on mathematical reasoning benchmarks--GSM8K and MATH.

**Weakness:**

1.  The empirical validation is less conclusive on multi-turn dialogue tasks. On the MT-bench-101 dataset, the proposed methods (OMPO and MPO) show mixed performance and do not consistently outperform baselines across several critical evaluation axes (e.g., GR, IC, PI). This suggests the method's pronounced advantages may be task-dependent and are not as universally applicable to open-ended dialogue as they are to structured reasoning.

**Audience:**

Yes

**Audience Explanation:**

Yes. These results should appeal to the TMLR readership, as multi-step alignment, optimistic mirror descent, and game-theoretic perspectives on RLHF are all active and rapidly developing research areas in machine learning theory and LLM alignment.

**Broader Impact Concerns:**

None.

**Claims And Evidence:**

Yes

**Claims Explanation:**

For the theoretical component, the authors provide solid proofs, and for the empirical component, they present corresponding experimental results.

**Requested Changes:**

1. In the second paragraph of Section 2.2, the statement
   “$ X_{h+1} \sim f(\cdot \mid S_h, A_h) $, which is equivalent to $ S_{h+1} \sim f(\cdot \mid S_h, A_h) $”
   is incorrect.

2. The citation for Definition 1 is missing.

3. The notation $ (\pi^{\*}, \pi^{\*}) $ is slightly misleading; the same concern applies to $ (d^{\*}, d^{\*}) $.

4. The manuscript does not clearly specify the exact relationship between $ \pi^{t+1} $ and $ \pi^{t} $ in Theorem 10.

5. Table 2 appears to omit the case of OMPO (iter = 1).

---

> ### Author Response · Authors · 2025-11-24
> **Response to reviewer za9D**
>
> We thank reviewer za9D for the thoughtful feedback, and the revisions are highlighted in blue in the revised version.
>
> > 1. statement on "equivalent to $S_{h+1} \sim f(\cdot|S_{h},A_{h})$.
>
> We delete the phrase "equivalent" and rephrase in the paper as follows:
> " For each step $h > 1$, a new action $A_h \sim \pi_h(\cdot|S_h)$ is sampled from the policy and the next prompt is sampled according to the transition function $f$, that is $X_{h+1} \sim f(\cdot|S_{h},A_{h})$. The next state is then constructed deterministically as $S_{h+1} = [S_h,\ A_h,\ X_{h+1}]$ by using the concatenation operator between sentences."
>
> > 2. Missing citation for Definition 1
>
> Our definition of an $\epsilon$-approximate Nash equilibrium follows the standard formulation used in the Markov game literature and online learning book. We have added the corresponding citations in the revised paper.
>
> > 3. Misleading $(\pi^\star, \pi^\star)$ and $(d^\star, d^\star)$.
>
> We added the following footnote in the paper to clarify:
> "In a two-player constant-sum Markov game, a Nash equilibrium is defined as $(\pi_1^\star, \pi_2^\star)=
>   \arg \max_{\pi}\min_{\pi'}\mathbb{E}_{S_1}
>      V^{\pi,\pi^\prime}(S_1,S_1),$ where $\pi_1^\star$ and $\pi_2^\star$ do not necessarily coincide in general.  However, in the symmetric game setting (our case), the two players face identical action and state spaces and share the same best-response structure.  In this case, the equilibrium policies coincide so that we write the equilibrium compactly as $(\pi^\star,\pi^\star)$. The same convention applies to the occupancy measures $(d^\star, d^\star)$."
>
>
> > 4. Does not clearly specify the exact relationship between  $\pi^{t+1}$ and $\pi^{t}$ in Theorem 10.
>
> For us the connection between $\pi^{t+1}$ and $\pi^{t}$ is specified by the main update in Theorem 10. We are happy to discuss this point further if the reviewer has suggestions on how to improve the presentation of that Theorem.
>
> > 5. Table 2 is missing values for OPMO(iter=1)
>
> For OMPO, we omit the result for iteration 1, because it is the same as MPO. OMPO relies on information from two adjacent steps, and at the initial step there is no previous step, so it coincides with MPO. We have added a sentence in the caption of Table 1.

---

> > ### Comment · Reviewer_za9D · 2025-11-28
> > **Acknowledgment of Response and Follow-up on Comment 4**
> >
> > Thank you for your detailed response and revisions. Most of my concerns have been addressed.
> >
> > I appreciate the authors' clarifications regarding comments 1, 2, 3, and 5.
> >
> > However, regarding comment 4, the manuscript still lacks an explicit formulation of the relationship between $\pi ^ {t}$ and $\pi ^ {t+1}$. The current text only describes a proportional relationship (i.e., using $\propto$). I suggest explicitly writing out the exact equality to improve clarity.

---

> > > ### Author Response · Authors · 2025-12-09
> > > **Response to comment 4**
> > >
> > > We thank the reviewer for the feedback. In the revised version, we provide the explicit equality for the policy update in Theorem 10. Specifically, we replace the proportional expression
> > >
> > > $$
> > > \pi^{t+1}_h(a|s) \propto \pi^{t}_h(a|s)\exp\Bigl(\tfrac{\beta}{H-h+1} Q^t_h(s,a)\Bigr)
> > > $$
> > >
> > > with the exact equality
> > > $$\pi^{t+1}_h(a|s) = \frac{\pi^{t}_h(a|s)\exp\Bigl(\tfrac{\beta}{H-h+1}Q^t_h(s,a)\Bigr)}{\sum\_{a' \in \mathcal{A}} \pi^{t}_h(a'|s)\exp\Bigl(\tfrac{\beta}{H-h+1} Q^t_h(s,a')\Bigr)},\quad  \forall s \in \mathcal{S},\  \forall a \in \mathcal{A}.$$

---

### Decision · Action_Editor_fDfb · 2025-12-19

**Recommendation:** Accept as is

**Additional Comments:**

The reviews have now settled into agreement. Reviewers find the paper technically sound, with claims that are supported by both the theoretical analysis and the experimental results. The issues raised in earlier rounds have been addressed in the revision, and no reviewer raises a remaining blocking concern.Reviewers repeatedly point to the same core contributions: framing multi-step preference alignment as a constant-sum Markov game, avoiding Bradley–Terry-style transitivity assumptions, and using optimistic mirror descent to obtain convergence guarantees that are not merely heuristic. The theory is viewed as careful and well presented. Earlier questions around notation, assumptions, and experimental setup have been clarified to the reviewers’ satisfaction. On the empirical side, dialogue results are mixed but reasonable, while performance on mathematical reasoning benchmarks is consistently strong. Reviewers consider this evidence adequate for TMLR and, in several cases, explicitly describe the paper as a solid archival contribution rather than a conference-oriented showcase. Overall, the paper meets TMLR’s standards for novelty, technical correctness, and relevance to its audience. I recommend acceptance.

**Audience:**

Yes

**Audience Explanation:**

See below.

**Claims And Evidence:**

Yes

**Claims Explanation:**

See below.

**Resubmission Of Major Revision:**

The authors may consider submitting a major revision at a later time.